# Advancing Sustainable Development: Emerging Factors and Futures for the Engineering Field

Grace Burleson [1,*], Jason Lajoie [2], Christopher Mabey [3], Patrick Sours [4], Jennifer Ventrella [5], Erin Peiffer [6], Emma Stine [7], Marie Stettler Kleine [8], Laura MacDonald [9], Jesse Austin-Breneman [10], Amy Javernick-Will [7], Amos Winter [11], Juan Lucena [8], David Knight [12], Scott Daniel [13], Evan Thomas [9], Christopher Mattson [3] and Iana Aranda [6,*]

1 Integrative Systems + Design, University of Michigan, Ann Arbor, MI 48104, USA
2 Department of English Language and Literature, University of Waterloo, Waterloo, ON N2L 3G1, Canada
3 Department of Mechanical Engineering, Brigham Young University, Provo, UT 84602, USA
4 Department of Food, Agricultural and Biological Engineering, The Ohio State University, Columbus, OH 43210, USA
5 Public and Urban Policy, New School, New York, NY 10011, USA
6 American Society of Mechanical Engineers, Engineering for Change, New York, NY 10011, USA
7 Civil, Environmental and Architectural Engineering, University of Colorado Boulder, Boulder, CO 80309, USA
8 Engineering, Design, and Society, Colorado School of Mines, Golden, CO 80401, USA
9 Mortenson Center in Global Engineering, University of Colorado Boulder, Boulder, CO 80309, USA
10 Department of Mechanical Engineering, University of Michigan, Ann Arbor, MI 48104, USA
11 Department of Mechanical Engineering, Massachusetts Institute of Technology, Cambridge, MA 02139, USA
12 Department of Engineering Education, Virginia Polytechnic Institute and State University, Blacksburg, VA 24061, USA
13 School of Professional Practice and Leadership, University of Technology Sydney, Ultimo, NSW 2007, Australia
* Correspondence: gburl@umich.edu (G.B.); arandai@asme.org (I.A.)

**Abstract:** This study set out to identify emerging trends in advancing engineering for sustainable development, supporting the engineering workforce to address wicked problems, and strengthening pathways between engineering education, industry, and policy. The following question guided this work: What are the emerging factors impacting the future of global sustainability efforts within engineering, and how can these be amplified to increase the impact of engineering for sustainable development? Using an adapted Delphi method with surveys, focus groups, and member-checking interviews, we hosted the American Society of Mechanical Engineers (ASME) 2022 Engineering Global Development (EGD) Stakeholder Summit. The summit convened industry leaders, innovators, and academics to explore emerging factors impacting the future of global sustainability efforts in engineering. This manuscript synthesizes emerging trends and proposes recommendations for engineering, particularly in the specific focus area of engineering for sustainable development (e.g., 'humanitarian engineering', 'global engineering'). Critical recommendations include the adoption of emerging cultural mindsets, which include: (1) take an interdisciplinary and multi-stakeholder approach, (2) consider dynamic and interconnected systems, (3) increase humility and intercultural competence, (4) prioritize diversity and inclusion, (5) increase localization and center community perspectives, (6) challenge the perception that engineering is neutral, and (7) broaden the goals of engineering. Ultimately, this study highlights pathways forward for the broader engineering community to more effectively contribute to advancing the United Nations Sustainable Development Goals.

**Keywords:** engineering; sustainability; global development; engineering education

## 1. Introduction and Background

Our global society is facing challenges of unprecedented scale, such as climate change, food insecurity, disease, access to safe and reliable water, and growing economic inequality.

Addressing such challenges will require interdisciplinary and transdisciplinary approaches from various backgrounds and fields. Engineers can play pivotal roles in addressing these challenges by reducing the progression of climate change, mitigating the impact of global emergencies, and advancing toward the United Nations (UN) 2030 Agenda for Sustainable Development. For example, the rapid innovation of health technologies, such as ventilators, vaccines, and protective equipment, which improved access to life-saving resources, highlights the impacts of engineers during the COVID-19 pandemic [1]. Engineers who can navigate across design, policy, and other fields are essential to addressing the most critical human development challenges, such as monitoring, adapting, and mitigating changing climates in agricultural and residential contexts, providing access to clean water, and developing affordable and accessible solutions to combat diseases such as HIV/AIDs, malaria, and coronavirus [2].

In recognition of these global challenges, efforts across various sectors and industries in the engineering field have sought to advance sustainability. The International Engineering Alliance defines engineering as a practice, body of knowledge, and set of techniques that can be purposefully applied to produce solutions of which the performance and impacts are predicted to the greatest extent possible [3]. Many engineering organizations are moving away from disciplinary and siloed definitions, for example, the American Society of Mechanical Engineers (ASME) defines their mission as one that "promotes the art, science and practice of multidisciplinary engineering and allied sciences around the globe" [4]. Engineering practice and knowledge can be applied to many different fields, including design, policy, business, and economics [5]. In this manuscript, we employ these broad definitions of engineering—focusing on the large global field of technical applied sciences [6].

More career paths for sustainability-focused engineers are becoming available within various organizations, such as consulting firms, social enterprises, non-government organizations, educational institutions, product manufacturers, international development organizations, government agencies, disaster response organizations, and corporate responsibility divisions [7]. Moreover, approaches such as the Environmental, Social, and Corporate Governance framework (ESG), which evaluates the extent to which an organization works on behalf of social and environmental goals, are incorporated into company practice [8]. While these efforts indicate progress toward broader sustainability goals, they remain dispersed and not widely adopted by the global engineering community. Insufficient efforts risk becoming a greenwashing tactic without robust accountability measures [9]. Other advances in industry include B-Corp certification [10] and the rise in presence and support for Social Innovation Enterprises [11].

In tandem with increasing sustainability efforts in industry, many academic institutions are establishing interdisciplinary departments and schools focused on sustainability and responsible innovation. One review identified over 85 academic institutions worldwide that provide learning opportunities for engineers interested in sustainable development; the foci of these programs range from community service and global development to peace engineering and social justice [12]. Importantly, these programs use a variety of philosophies and different terminology to name their fields of study, including 'humanitarian engineering', 'global engineering', 'engineering for community development', 'rural engineering', and 'social innovation', among others [13]. Programs of this nature have contributed to developing socio-technical skill sets engineers need to address the complexity of sustainability challenges [14]. In 2021, a large interdisciplinary working group co-developed 15 learning objectives for global engineering graduate programs that emphasized ethics, cross-cultural humility, complex systems analysis, and climate resilience, among others [5].

Despite these efforts to address sustainability goals, the global engineering profession, including academic, industrial, and public systems, is unprepared to tackle these significant socio-technical challenges. The engineering industry lacks guiding principles, incentives, and tools to make sustainably focused decisions. While the UN Sustainable Development Goals (SDGs) provide a "North Star" [15], the framework lacks a specific roadmap or set

of operating instructions. Many enterprises strive to meet market demand and decrease their carbon footprint but encounter operating and business roadblocks, among many other socially, environmentally, and economically difficult decisions. While efforts such as Engineering for Social Responsibility [16] and Social Justice [17], as well as frameworks such as Engineering for One Planet [18], can highlight these challenging decisions, there remains a need for cohesion around these guiding principles. Opportunities are growing, yet challenges persist amongst junior engineers seeking career pathways and opportunities that focus on sustainability. Unfortunately, students in engineering for global development and related fields worry about the difficulty of securing a career in this area instead of a career in other engineering fields [19].

From a training and academic perspective, an ongoing re-examination of engineering education has identified the need for engineers to think and problem-solve in new ways [5]. Most engineering curricula do not adequately prepare technical talent to address what some call 'wicked sustainability problems', e.g., climate change, poverty, and resource scarcity [20]. These challenges are complex, do not have clear boundaries, and are driven by stakeholders' differing values, interests, and conceptions of the problem and potential solution [21]. To better equip the science, technology, engineering, and math (STEM) workforce to tackle wicked problems, students need to link multidisciplinary perspectives from the social sciences to critical design skills [22]. In addition, developing intercultural competencies with strong interpersonal and professional skills will be critical to challenging the complexities of these issues from a wide range of perspectives and scales [23]. Currently, curricula emphasize technical skills, training students to reduce large complex systems to simpler quantitative models. While modeling is a critical component of large-scale problem-solving, many curricula emphasize the technical at the expense of other necessary skills, such as interdisciplinarity, ethics, and cross-cultural competence [24]. Further, since considerable aspects of our engineering training, design, and practice are determined by WEIRD (Western Educated Industrial Rich Democratic) ideals and perspectives [25], with a few notable exceptions, such as the Engineers Without Borders Australia Design Summit, current STEM education largely lacks meaningful cross-cultural experiential learning [26].

As engineering education must be updated to train the next generation, the engineering profession must become more inclusive, cooperative, and socially and environmentally responsible [15]. Globally, representation within the engineering field remains inequitable. For example, in the United States, Black engineers make up only 5% of the engineering workforce, while women make up just 15%; these groups are well below their total respective workforce shares of 11% and 47% [27]. While prior literature has found that humanitarian work attracts women and minorities to engineering [28,29], other studies have found that women pursuing more socially engaged engineering were especially at risk of leaving the engineering profession due to discontent with their experiences [30]. Contributions from a broad array of lived experiences and perspectives are needed to collectively address the ever-growing sustainability challenges our society faces.

To support the engineering workforce to address these "wicked problems", the Engineering Global Development committee at the American Society of Mechanical Engineers (ASME) hosted a Stakeholder Summit to strengthen pathways between engineering education, industry, and policy. The ultimate goal was to convene innovators, academics, NGOs, and other leading voices in engineering whose work advances sustainable development for a high-level exploration of the emerging factors that impact (and will ultimately shape) the future of the engineering profession. Among ASME's recommendations was the development of an umbrella term, "Engineering for Sustainable Development" (ESD), to identify various focus areas actively contributing to sustainable development efforts (including Engineering for Global Development, Global Engineering, Humanitarian Engineering, and Social Innovation, among many others). As such, this paper uses the umbrella term, 'ESD', which refers to the broad interdisciplinary practice of engineering to improve the quality of life of society and the environment worldwide, mainly related to the UN Sustainable Development Goals (SDGs). Ultimately, this paper presents the summit's findings that

highlight emerging and recommended cultural mindset shifts and proposes pathways for the engineering community to advance the Sustainable Development Goals.

## 2. Methodology

### 2.1. Engineering Global Development at ASME

ASME, a long-standing not-for-profit membership organization, enables collaboration, knowledge sharing, and skills development across all engineering disciplines and continents to advance engineering for the benefit of humanity [31]. ASME's Engineering Global Development (EGD) team focuses on building engineering capacity and talent around the world to solve urgent, local, and global challenges, including access to clean water, adequate sanitation, housing, reliable electricity, transportation, food production, accessible education, health care, information, communications, and more. Through partnerships, training, resources, and platforms that accelerate the development of long-term solutions for vulnerable populations, ASME supports the advancement of the UN Sustainable Development Goals (SDGs) through a global network at Engineering for Change (E4C), applied Impact Projects, workforce development programs (e.g., the E4C Fellowship) and codified engineering knowledge (e.g., the E4C Solutions Library). These EGD platforms connect and catalyze the technical workforce for good as engineers increasingly seek to use their skills to create positive social and environmental impact.

### 2.2. Overall Study Design

To advance engineering for sustainable development, support the engineering workforce to address many of the wicked problems highlighted above, and strengthen pathways between engineering education, industry, and policy, we set out to explore existing trends and future pathways for the engineering community to improve its contributions to advancing the Sustainable Development Goals, focusing both on engineering practice and education. The following question guided this work: What are the emerging factors impacting the future of global sustainability efforts within engineering, and how can these be amplified to increase the impact of engineering for sustainable development?

#### 2.2.1. Study Objectives

The following objectives guided the design of this study:

1.  To describe primary graduate and professional competencies for engineers, which are valued by employers pursuing sustainable global development.
2.  To describe the professional pathways currently available for engineers trained in sustainable global development.
3.  To understand the current and shifting global drivers on the professional pathways (e.g., inequity, poverty, pandemic, social justice, climate change).
4.  To identify opportunities for the engineering for sustainable development sector to increase its positive impact on society while increasing the availability of long-term and en masse professional pathways within engineering for sustainable development.
5.  To suggest opportunities for ASME to advance sustainable economic, social, and environmental global development objectives.

#### 2.2.2. Overall Methodology: Adapted Delphi Method

To address our research question, we used an adapted Delphi method to gain insights from a select group of experts. The Delphi method is characterized as a "method for structuring a group communication process so that the process is effective in allowing a group of individuals, as a whole, to deal with a complex problem" [32]. Delphi methods have been applied to identify trends and perspectives in engineering fields, such as the construction management [33,34], environmental health [35], and transportation [36] fields. We designed our study to use an adapted Delphi method to suit the objectives outlined in Section 2.2.1, which included conducting surveys, a one-time summit with facilitated focus groups, and

iterative *member checking* with participants, i.e., validation of results through feedback with participants [37,38]. Figure 1 presents a summary of the methodology used in this study.

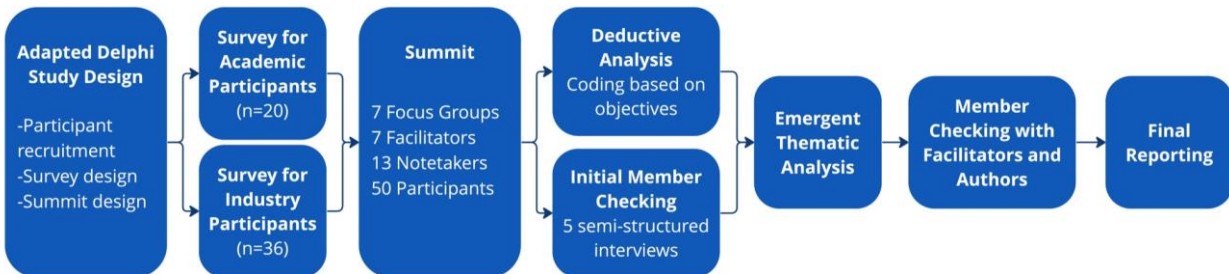

**Figure 1.** Methodology process summary.

*2.3. Stakeholder Recruitment*

We recruited stakeholders across private industrial, academic, and public sectors to ensure a diverse and wide range of expertise and perspectives. Stakeholders were recruited from engineering organizations and employers with existing sustainability strategies or practices in place that were connected to ASME and EGD, specifically. The ultimate goal was to generate groups of individuals with diverse, cross-sector, and multidisciplinary experience in engineering for sustainable development.

Overall, eight facilitators, 13 notetakers, and 55 participants (50 participants in the summit focus groups and five in member-checking interviews, described in Section 2.4) were involved in primary data collection during focus groups. Participants joined from all over the world, including Canada, India, Italy, Kenya, and the United Kingdom, with the majority participating from the United States. Participant background and credentials are summarized in Table 1.

**Table 1.** Summary of participants' background and credentials.

| Industry Respondents | |
| --- | --- |
| Industries represented by participants | Utilities, construction, education, financial services, telecommunications, health services, oil and gas, public service, manufacturing, product design, philanthropy, international development, medical devices, software |
| Roles in organization represented | Program officer/manager, project manager, mechanical engineering, vice president (e.g., of sustainability), CEO, executive chair, executive/managing director, consultant, president, chief engineer, |
| Size of organizations represented | Large (250+ employees), Medium-sized enterprise (50–249 employees), Small enterprise (10–49 employees), Microenterprise (1–9 employees) |
| **Academic Participants** | |
| Primary disciplines represented | Mechanical engineering, civil engineering, engineering education, global/humanitarian engineering, bioengineering, environmental engineering, systems engineering |
| Roles in academia | Assistant professor, associate professor, professor, associate dean, assistant dean, director, research staff |

*2.4. Data Collection*

2.4.1. Pre-Summit Surveys

Before the summit, we conducted two surveys: one for stakeholders affiliated with academic institutions and one for industry more broadly. The two surveys were developed in consultation with university researchers familiar with such methodologies, incorporating research-based survey design practices [39]. Both surveys focused on collecting perspectives and values of engineering skills, training, and career pathways related to engineering for sustainable development. The primary goal of each survey was to collect responses to incorporate into the design of the summit to prime participants and better elicit their reflections during the summit. Within each survey, specific terms were carefully selected based on language and terminology that participants in each sample were familiar with. For example, we used 'engineering for global development (EGD)' in the academic participant survey because the ASME academic community uses this terminology. In contrast, we used 'sustainable development' in the industry participant survey because this aligns with their commonly used terminology. The survey outlines and some example questions are presented in Table 2, and the complete surveys are presented in Appendix A.

**Table 2.** Overview of ASME EGD Stakeholder Summit pre-surveys for participants with some example questions.

| Survey Sections | Industry Participants | Academic Participants |
|---|---|---|
| Participant background | <ul><li>Industry domain</li><li>Role in your organization</li><li>Size of organization</li><li>Engineering disciplines in your organization</li></ul> | <ul><li>University affiliation</li><li>Primary academic discipline</li><li>Position/title</li><li>What opportunities do you specifically lead/mentor to provide EGD-related training for students?</li></ul> |
| Perspectives and values | <ul><li>How much do you agree/disagree with the following statement? "Engineers trained in sustainable development are valuable to our organization"</li><li>Why is your organization interested in hiring engineers trained in sustainable development?</li></ul> | <ul><li>How much do you agree/disagree with the following statement: "Training engineers to work in EGD-related careers is core to my teaching philosophy"</li></ul> |
| Engineering skills and career pathways | <ul><li>What are the top 3 skills that you most value in your ideal engineering candidates with experience in sustainable development?</li><li>What are some career paths for engineers to practice sustainable development in your organization?</li></ul> | <ul><li>What are the top 3 skills that you train your students preparing for EGD-related careers?</li><li>In your experience, what types of learning opportunities provide students with the best training/preparation for EGD-related careers?</li></ul> |
| Future-casting | <ul><li>How much do you agree/disagree with the following statement: "When considering how the practice of engineering will change in response to the Sustainable Development Goals, I feel my organization is well prepared".</li><li>What factors are enabling these pathways for engineers to practice sustainable development (now or in the future)?</li></ul> | <ul><li>What are some EGD-related career paths for engineers that you mentor students towards now or in the future?</li><li>What are you finding to be enablers for your students to pursue EGD-related career paths?</li></ul> |

2.4.2. Facilitated Focus Groups

During the 3 h summit, we conducted two focus-group sessions. Each session involved seven groups with 5–8 participants, a facilitator, and 1–2 notetakers. Using a virtual whiteboard platform all participants had access to throughout the session, facilitators led participants through discussion-based activities driven by the various session goals shown in Table 3. Survey results were presented and discussed to encourage reflection. Appendix B presents some examples of how content was presented during the sessions.

**Table 3.** Focus-group sessions and topics.

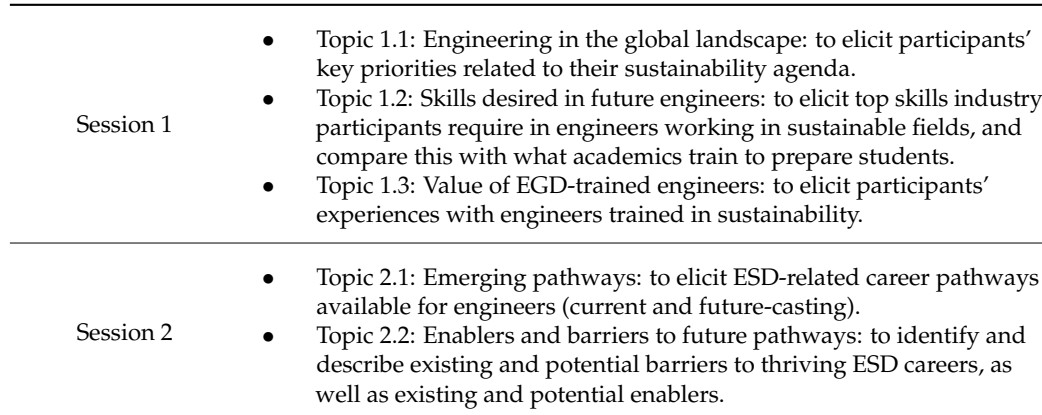

| Session 1 | • Topic 1.1: Engineering in the global landscape: to elicit participants' key priorities related to their sustainability agenda.<br>• Topic 1.2: Skills desired in future engineers: to elicit top skills industry participants require in engineers working in sustainable fields, and compare this with what academics train to prepare students.<br>• Topic 1.3: Value of EGD-trained engineers: to elicit participants' experiences with engineers trained in sustainability. |
|---|---|
| Session 2 | • Topic 2.1: Emerging pathways: to elicit ESD-related career pathways available for engineers (current and future-casting).<br>• Topic 2.2: Enablers and barriers to future pathways: to identify and describe existing and potential barriers to thriving ESD careers, as well as existing and potential enablers. |

All facilitating materials were prepared in advance so that the topics and questions for the groups would follow the same structure, and facilitators all participated in training sessions before the summit. Although facilitators were instructed to follow the topics and recommended durations for each activity, they were also encouraged to allow the discussion to flow based on the expertise and experiences shared in their specific group. Data were collected in two ways: virtual sticky notes, comments written by participants on the virtual whiteboards, and detailed notes drafted by notetakers in each group. Appendix C presents a brief example of notes drafted and virtual sticky notes collected, which were the primary base of data analysis.

*2.5. Data Analysis and Member Checking*

Our data analysis was grounded in the five objectives presented above. First, notes from the summit and virtual whiteboards were examined independently by two individual evaluators, and participants' responses were grouped (i.e., deductively coded) into seven categories that mapped themes driven by our objectives: (1) Barriers for ESD careers, (2) Enablers for ESD careers, (3) Career pathway opportunities, (4) Skill development in ESD-trained engineers, (5) Shifting career pathways, (6) Changing mindsets, and (7) Other. For each of the seven focus groups, excerpts within each theme from the two evaluators were consolidated into a single document and organized by emerging themes (e.g., changing mindset: humility, shifting career pathways: policy careers). Next, three authors conducted semi-structured interviews with five key stakeholders who could not attend the summit for initial member checking and to assess the resonance and transferability of the preliminary results—between two to three authors conducted each of the interviews. Interview protocols followed the structure presented in Table 4.

Next, we evaluated and synthesized the complete set of clustered data (i.e., all seven documents with consolidated coding) to identify emergent findings, including current trends, potential pathways forward, and recommendations for the engineering field. Using a shared word processing tool, we incorporated data from the member-checking interviews with stakeholders to further fine-tune the claims being developed. Following recommendations by qualitative research scholars [40], most of the authors participated in group discussions to negotiate the interactions between the raw data, emerging themes, and

the overarching objectives. Findings (i.e., the developed claims and themes) within each objective were reviewed, discussed, and iterated as part of a reflexive thematic analysis performed by the authors.

**Table 4.** Interview protocol structure with key questions for member checking with additional stakeholders.

| | |
|---|---|
| Introduction | • Interviewers present a summary of the ASME EGD Stakeholder Summit, including the overall goals and main session topics |
| Career pathways | • What skills and experiences are you needing in your organization? Any new positions developing?<br>• What barriers or enablers to sustainability careers do you see? |
| Trends at large | • What is needed for the sector to progress and achieve sustainability goals? |

The themes and claims within objectives 1–3 were then organized into four high-level categories to serve as the organization for reporting results in Section 3: 'Global drivers lead to mindset shifts', 'Increased availability of professional pathways for ESD engineers', 'Broader skill sets for ESD engineers are valued', and 'Gap between industry needs and educational outcomes.' When reporting the results, we carefully chose key findings from survey data or exemplary excerpts from our qualitative data to illustrate our analysis process and provide nuance to the findings [41]. All presented excerpts are taken from the notetakers' detailed participation notes. To develop the discussion section (Section 4), we organized the key themes and claims identified for objective 4 to suggest opportunities for the field at large, which we organized into the following subsections: 'Engineering for Sustainable Development: Terminology Recommendations', 'Emerging and recommended cultural shifts for field at large', 'Incentives, metrics, and decision-making frameworks', and 'Engineering education'. Findings for objective 5 are underemphasized in this report and are being used to inform future ASME plans and programs.

## 3. Findings: Trends in Engineering for Sustainable Development

### 3.1. Global Drivers Lead to Mindset Shifts

During the summit, participants affirmed that across the field of engineering, there is an increasing awareness of social and environmental impacts of innovation, which the climate crisis, COVID-19 pandemic, and broader global social justice movements around the world have amplified. What used to be considered 'problems' in primarily emerging markets, such as the social impact of technology, resource allocation, and supply chain issues, were described as 'front-and-center' priorities for all markets. Participants referenced how the UN Millennium Development Goals (the predecessors to the SDGs) were previously only for 'developing nations' (a term that was very ill-defined), while the SDGs are now for all member nations. Participants described the lines between so-called 'developed' and 'developing' nations as increasingly 'blurred' due to the growing awareness and mitigation of historical colonial approaches, 'parachute aid', and 'voluntourism'. There was palpable advocacy for engineering to move away from the globalization of western engineering towards more localized global engineering, e.g., building local capacity and respecting local expertise and knowledge to attend to social justice considerations and historical inequities. For example, one participant, who represented an incubation firm that supports social innovation start-ups, described:

> *We have more focus on grassroots innovation and impact delivery in local ecosystems. We are prioritizing building more local capacity to support these initiatives.*

Participants stressed how the awareness of environmental and social considerations is rapidly growing in the public eye, particularly among younger generations who want to contribute to and consume products and services that are more equitable and sustainable.

This demand has led to increased corporate social responsibility (CSR) initiatives, standards, and certifications, such as B Corp, Cradle to Cradle Certified, and other various fair trade and responsibly sourced certifications. Increased public engagement with the social and environmental impacts of innovation has led to efforts to establish methods to develop and measure these impacts. Metrics, such as the Environmental, Social, and Governance (ESG) criteria, enable investors to screen initiatives and make socially conscious decisions while attracting a workforce that values social and environmental responsibility. For example, one participant from an engineering consulting company claimed:

> *Companies must include and demonstrate commitment to ESG to attract and retain talent.*

Notably, many participants highlighted the importance of raising awareness for all engineers to think more deeply about the environmental impact of their work. For example, a participant representing a foundation that supports sustainability initiatives claimed:

> *The Intergovernmental Panel on Climate Change (IPCC) showed us how fast the climate was changing. Engineers are and should have a more holistic view. What is the impact of their products? What are they developing? This is a high priority of our organization, to get engineers to understand these impacts as they are developing, designing, manufacturing, and disposing of their products—this also needs to get into the curriculum at universities.*

Given the increasing interest and need for social considerations, such as Diversity, Equity, and Inclusion (DEI), many engineering companies invest in DEI metrics and reporting. However, these initiatives are not widespread or standardized, which some participants claimed can result in 'impact washing', defined as marketing claims of social or economic impact without sufficient evidence [42]. Participants highlighted the importance of DEI initiatives to attract and retain the diverse workforce necessary for tackling these considerable sustainability challenges. For example, one participant claimed:

> *This is an area where our field struggles immensely. If you don't leverage your diverse workforce, they aren't going to stick around . . . Companies are struggling to implement true DEI programs. If they fail to use people's unique skills, they will leave. They need to create a sense of belonging within their companies.*

Advancements in engineering product and service design methodologies, such as inclusive, participatory, and contextual design, provide recommendations for engaging stakeholders ethically and effectively throughout design and innovation processes. For example, one participant representing an organization that hosts forums to convene engineers who develop initiatives for broader change claimed:

> *We have power in shaping the world as technology is created, but it needs to be an inclusive process.*

Another participant, representing a large international non-governmental engineering organization, highlighted the increase in demand for socially engaged design processes, claiming:

> *A lot more training and demand for human-centered design is happening at our organization. You need new ways to ensure you are incorporating the users' experience, needs, preferences, capacities . . . and that development is collaborative and requires empathy, and to move us away from top-down approaches and perspectives.*

Government-enacted regulations push companies to reduce their environmental impacts, such as the U.S.'s Corporate Average Fuel Economy (CAFE) and the Directive on Single-Use Plastics in the European Union. Discussing the case of a governmental ban on sulfur-rich coal in Mongolia in 2019, one participant emphasized how government regulation helped subsidize the development of alternative energy sources, claiming:

> *You need both guidance and enforcement for the uptake of sustainability to work well. Where there's obvious financial benefit, like waste to energy, that can drive itself forward,*

*and the more of that, the better. On the other side, where what you're doing is, say, creating an absence of pollution, that's harder for the market to drive, and that's where we need someone to come in and regulate.*

How best to balance regulation and guidance was a recurring topic, with regulation seen as valuable in some contexts and guidance as useful in others. One participant found that successful regulations occurred when influential organizations "made rules that are outcome rather than process based. Not 'do x, y, z' but 'achieve x, y, z'". When it comes to navigating the "minefield" of accreditation and certification, another participant felt that "rather than regulatory, there is a need for a guidance role" to help companies. Environmental metrics and policies are needed to drive these changes; a participant shared:

*Companies take the concept of risk very seriously, so without the science about technical performance characteristics of sustainable materials and how to integrate these materials into their offerings–like there already are with concrete or steel—then actually making more sustainable choices can be difficult, especially to stand up and justify to the client and rest of the team, so you fall back on the familiar. But, if there are examples that challenge that perception, and show that it's not going to cost more money, in the long run, to be inclusive and sustainable, then companies are more open to adopting them.*

There is even less unification of performance metrics for addressing and predicting the social impacts of innovation. For example, one participant described how difficult it could be to select and prioritize specific social metrics over others since they are highly interrelated. However, participants were clear that identifying social metrics was more difficult than environmental metrics. For example, one academic participant claimed:

*Assessment has been focused on environmental parameters that can be measured, but there are limitations when we try to assess human and social life cycles. They are difficult to quantify. What are designs doing to emotional health? And, are we missing out on the voices of underserved populations?*

### 3.2. Professional Pathways Are Becoming More Available to ESD Engineers

A variety of ESD career opportunities exist, including (1) technology and design, (2) product and project management, (3) supply chain management, (4) financing and funding decision making, (5) social entrepreneurship, (6) impact reporting, and (7) academic pathways. Of these, the most commonly mentioned ESD career opportunities available were the design and development of sustainable processes, products, and programs. Next, participants cited opportunities in impact reporting, such as measuring progress and outcomes in sustainable development and project management. In general, there was great optimism about increasing opportunities to practice sustainable development, with more than 50% of participants claiming that the number of opportunities in their organization will strongly increase in the next three to five years, see Figure 2.

Participants also emphasized an increasing presence of engineers in historically non-traditional career pathways, such as those in finance and policy. Engineers with technical and contextual expertise play roles in funding decision making, policy development, and application. One participant suggested that many sustainable development initiatives could be "reimagined with more engineers at the table" in financial management roles. In addition to formalized career pathways, summit participants argued that engineers specializing in ESD should advocate for social and environmental accountability of the technology and engineering industry. Other participants suggested that engineers should be formally engaged in supporting policy and described some current opportunities to do so:

*There is a program to get engineers into the policy realm. Policy could be a big enabler . . . The AAAS [American Association for Advancement of Science] Fellowship is an example; people with a PhD go to a federal agency to try to translate their understanding into policy, a great experience to have early in their career.*

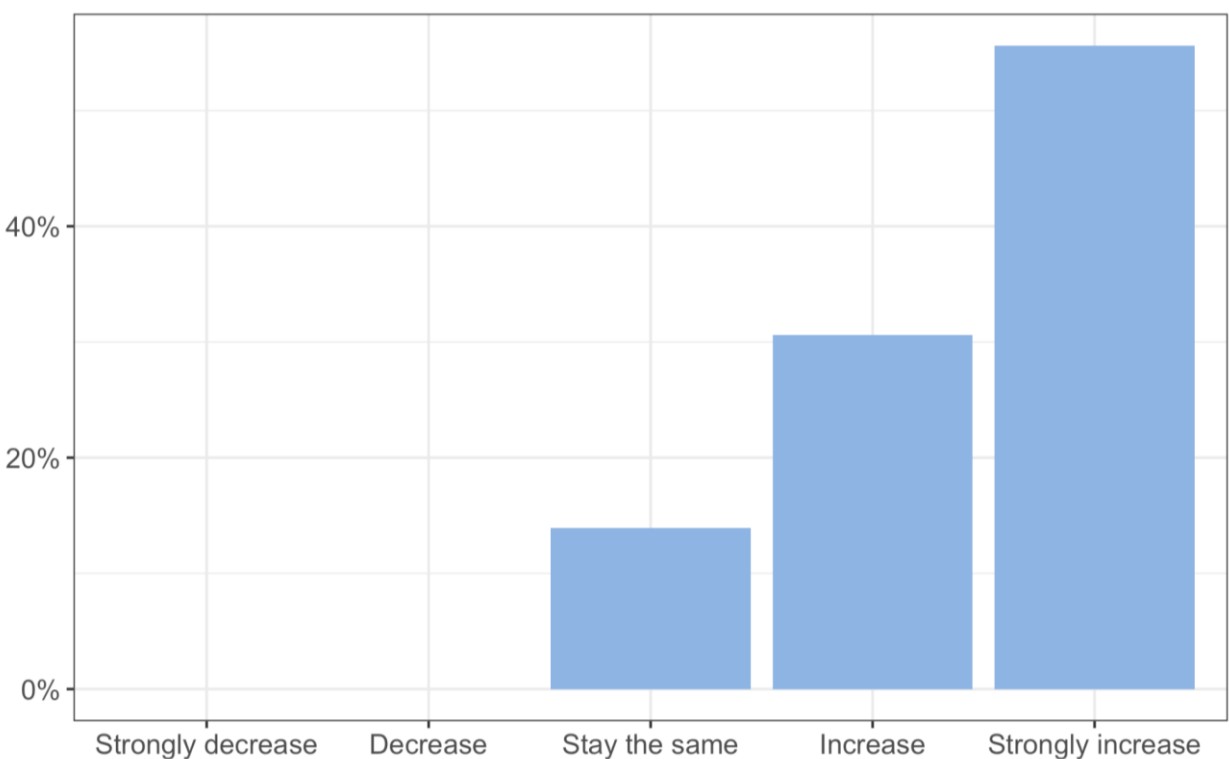

**Figure 2.** Industry survey responses to the question "Within the next three to five years, opportunities to practice sustainable development within our organization are likely to . . . ".

Engineers play a role in applying existing metrics to company financial and design models. One participant described their role in developing funding requirements to ensure that their organization selected proposals aligned with recommended practices and broader social and environmental goals. Specifically, by developing funding requirements, she provided incentives for development organizations to incorporate specific equity considerations. Within academia, there has been a modest increase in engineering faculty positions for researchers focusing on sustainability and social impact. For example, within the last two years, the Mechanical Engineering Department at the University of Michigan hired four tenure-track faculty members whose research focused on "engineering for social justice".

Epistemologies and priorities for ESD vary widely across various career paths and institutions. They are influenced by disciplinary perspectives and standards of work, applications of empathy and humility, social justice principles, and differences in engineering-specific technical expertise and knowledge domains. Some of the opportunities for engineers interested in global sustainability career paths identified by summit participants include:

- Sustainability Manager: Managing various sustainability programs and environmental impact reporting, including managing and implementing institutional climate action plans.
- Consultants: Working directly with companies to develop climate action plans, reporting structures, and programs to enhance sustainability outcomes.
- Technical Associates: Providing technical expertise on projects, particularly for non-profit organizations.
- Social Innovators & Start-ups: Undertaking entrepreneurial projects to develop a scalable business model focused on improving environmental and/or social well-being.

Educational and training opportunities for ESD careers are rising, and more engineering students are interested in pursuing them. ESD is increasingly taught through extra-curricular, certificate, minor, major, and graduate degree programs, which are quickly

expanding across the globe through various pedagogical, curricular, and philosophical approaches. However, this has introduced specialized vocabulary that can be difficult to connect with the practice and education of engineering. One participant described concerns with different terminologies used across different disciplines:

> *Sometimes we talk about the same things but use different words, and use of language can be exclusive. Maybe we need a taxonomy of technical terms. Does that mean hiring people from different disciplines at [the] leadership level? Or, is it creating a level of empathy amongst personnel so they can appreciate where their expertise ends and know what expertise can round them out?*

The study revealed a gap between how academia and industry prepare graduates for ESD careers. Participants noted that many students and early-career engineers mistakenly assume ESD work only occurs in Low- and Middle-Income Countries. Many opportunities exist for ESD work across all countries and contexts, regardless of national income level. Organizations offer fellowships for students to connect them to ESD careers with valuable work experiences, such as the ASME Engineering for Change Fellowship and Tech Stewardship Program. As such, many participants felt that ESD pedagogy should expand students' understanding of ESD career opportunities.

Additionally, participants advocated for a need to develop and retain a diverse engineering workforce. Engineers from disadvantaged backgrounds are diverted from ESD career paths due to the lack of high-paying positions compared to traditional engineering roles. Indeed, while ESD-related work attracts diverse talent due to its breadth of applications, there are many opportunities to improve retention through more explicit, more inclusive career pathways and paid opportunities. A participant shared:

> *Paid internships for students are important for competition and elevation, but there are too many unpaid opportunities within engineering for global development, leading to issues of access.*

Diversifying the engineering workforce and strengthening the impact of ESD requires recognizing power dynamics, cultural differences, implicit biases, and privilege within engineering work and dismantling the historical divide between 'international' staff and 'local' staff in ESD. Individuals among the 'international' staff, who carry privileged social constructs, must reflect on their identity and motivations for working in the ESD sector. 'Local' individuals, who carry variations of historically marginalized identities, and hold social, navigational, technical, linguistic, and cultural capacities instrumental in leading global reform efforts and reaching sustainable development targets, should be fairly compensated and hold decision-making roles.

Further, participants stressed the need for diverse teams of engineers to address problems and reduce systemic biases. One participant cited Caroline Criado Perez's *Invisible Women: Data Bias in a World Designed for Men* as an example of how historical inequities and implicit bias in engineering solutions can result in products that discriminate based on gender [43].

### 3.3. Broader Skill Sets for Engineers Are Valued

While engineers are known for their technical skills, participants in the study stressed the importance of developing engineers' interpersonal and reflexive skills, especially in leadership and cross-cultural collaboration. Traditionally, these skills have been treated as adjacent to technical competencies. However, as Figure 3 shows, increasing importance is placed on teaching that inculcates fundamental skills and technical competencies as complementary to one another and co-constitutive of professional engineering practice. Participants suggested that infusing broader personal and professional skills within existing technical education, especially in ways that seek to resolve traditionally imposed demarcations between them, would also allow engineering educators to address these competencies while still meeting the specific accreditation demands of engineering degrees.

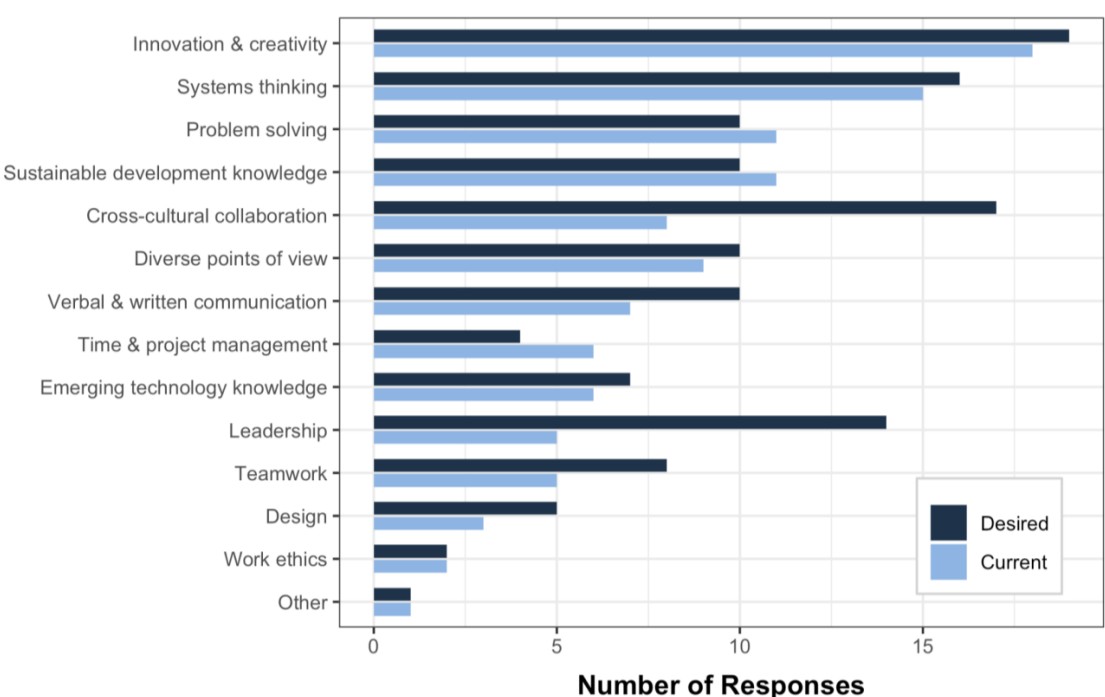

**Figure 3.** Top skills of engineering workforce trained in ESD. Participants from industry (n = 36) were each given the option to select their top three skills.

Despite recent efforts to address variations in learning objectives through working groups and coalitions (e.g., the Humanitarian Engineering Community of Practice [44]), university education must unify efforts and incentives for more robust ESD training. Participants suggested that engineers draw from fields like Science and Technology Studies (STS), Philosophy of Technology, and other social sciences, to strengthen ethical approaches to engineering solutions and critical decision-making skills. For example, some suggested that curricula recommendations through the Accreditation Board for Engineering and Technology (ABET) in the U.S. and the Engineers Australia Stage 1 Competencies in Australia could provide guidelines and incentives for educators to incorporate ESD training.

To better prepare for the collaborative and connected nature of Industry 4.0, which is characterized by rapid changes to technology, industry, and society due to increasing interconnectivity and intelligent automation, engineers must also be equipped with broader systems-thinking skills, such as defining problems systematically to devise appropriate technical solutions better. Our survey results showed that within industry, there is a gap between the current and desired skills of engineers trained in ESD concerning leadership and cross-cultural collaboration. Interestingly, while educators prioritized cross-cultural collaboration as among the top three skills they train in students, only one academic respondent cited leadership as a top three skill, which is likely because opportunities for students to gain leadership experience are primarily found in extra-curricular activities, which has been problematized in prior work [45]. This result suggests a mismatch between the degree of leadership instruction in engineering programs and its demonstration among early career engineers. Future research could explore this question of leadership in ESD-related careers and how students can develop and demonstrate it, especially within programs focused on fostering collaboration and humility.

### 3.4. Gap between Industry Needs and Educational Outcomes

Across our participants, ESD education was described as crucial for the training and preparation of all future engineers. The skills highlighted above are critical for any practitioner working on technological solutions since these require consideration of resource allocation, product lifecycle, and social impacts and are increasingly developed

on dispersed global teams. Understanding key drivers of innovation and development is especially critical for developing this holistic approach in which engineers can analyze problems and then develop sustainable and equitable solutions that consider the broader socio-cultural and environmental contexts in which they will be embedded. When asked how prepared their organization was to respond to the Sustainable Development Goals, less than 20% of respondents strongly agreed that their organization was ready, see Figure 4. Survey responses also revealed differences between available career opportunities and the paths academia is mentoring students toward, see Figure 5.

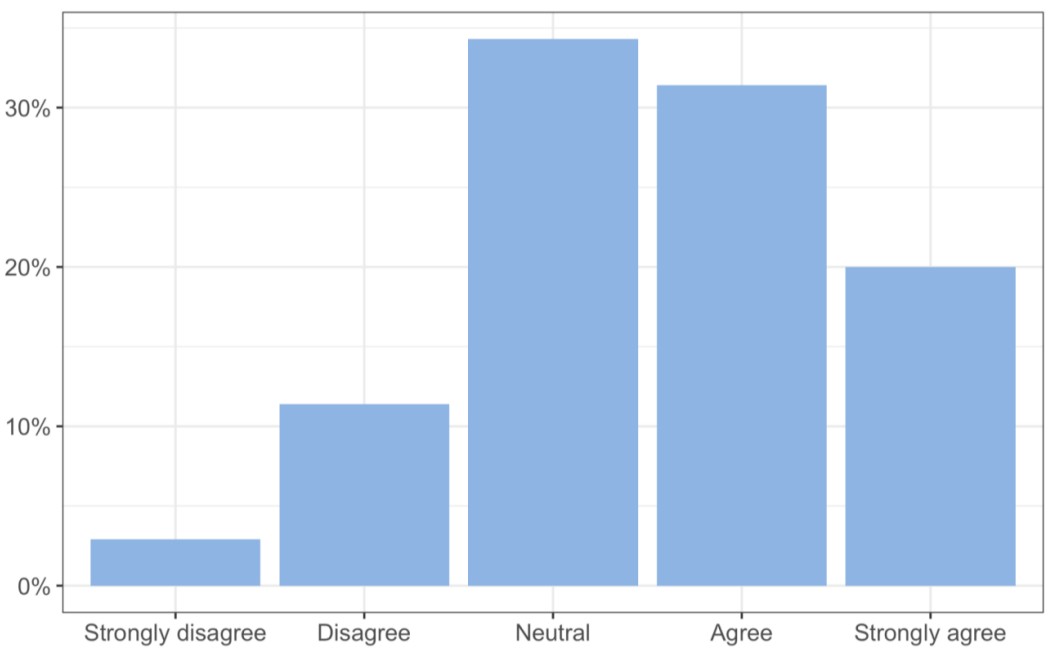

**Figure 4.** Responses to the question: "How much do you agree with the following statement? When considering how the practice of engineering will change in response to the Sustainable Development Goals, I feel my organization is well prepared".

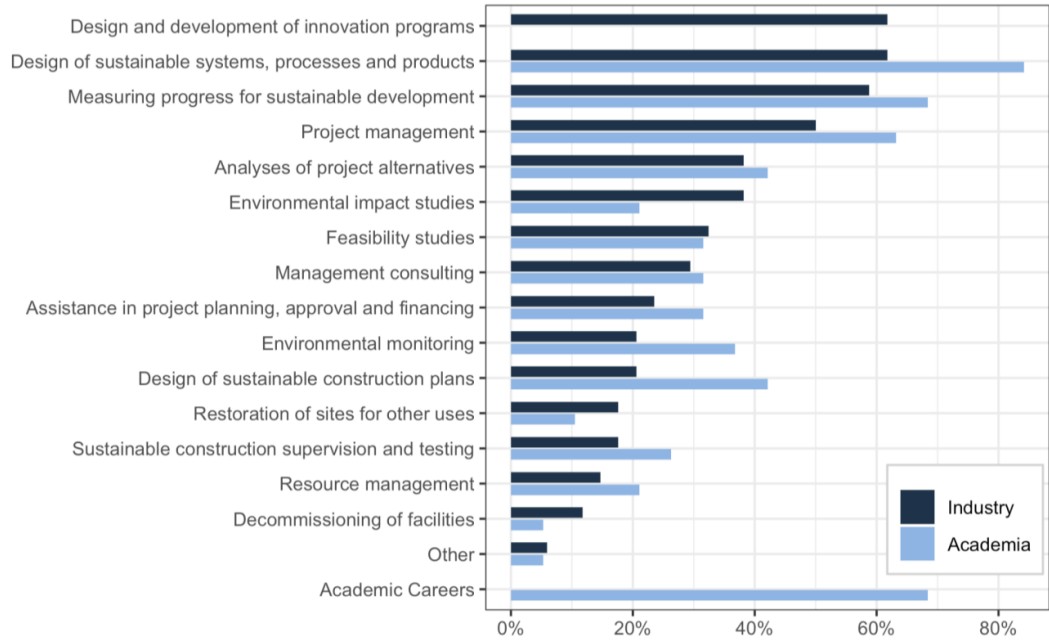

**Figure 5.** ESD career pathways identified by industry respondents and career pathways academia mentors students toward.

## 4. Discussion

Findings from the summit, including emergent themes, were distilled into recommendations for action. The subsections below outline these pathways in detail, including recommendations for cohering terminology (Section 4.1); shifting engineering cultures (Section 4.2); developing incentives, metrics, and decision-making frameworks (Section 4.3); and transforming engineering education (Section 4.4).

### 4.1. Engineering for Sustainable Development: Terminology Recommendation

Due to the expansive nature of the fields of engineering that aim to achieve sustainability and social equity goals, findings from this study have led ASME to pursue the use of an umbrella term: 'Engineering for sustainable development' as a way to name the specialty while acknowledging that there are similarities and differences with areas within. Using agreed-upon and codified terminology should enable more coordination across ESD-related efforts and increase its recognition within engineering. Engineering and related sectors have multiple terms for work focused on ESD; see Figure 6. It is critical to note that different terminology for ESD-related fields of study, such as Engineering for Global Development [46], Global Engineering [47], Development Engineering [48,49], Humanitarian Engineering [13,50], Design Justice [51], and Activist Engineering [52], among many others, cannot necessarily be used interchangeably, as these represent different schools of thought and academic ancestry (e.g., "Humanitarian Engineering" has received official sanction in Australia by the peak research funding agency, the Australian Research Council, and by the peak body for professional engineers, Engineers Australia). However, a single umbrella term and accompanying lexicon or terminology acceptable to and adopted by the field as a whole will reduce confusion, facilitate coordination, and lay the groundwork for sustainable efforts.

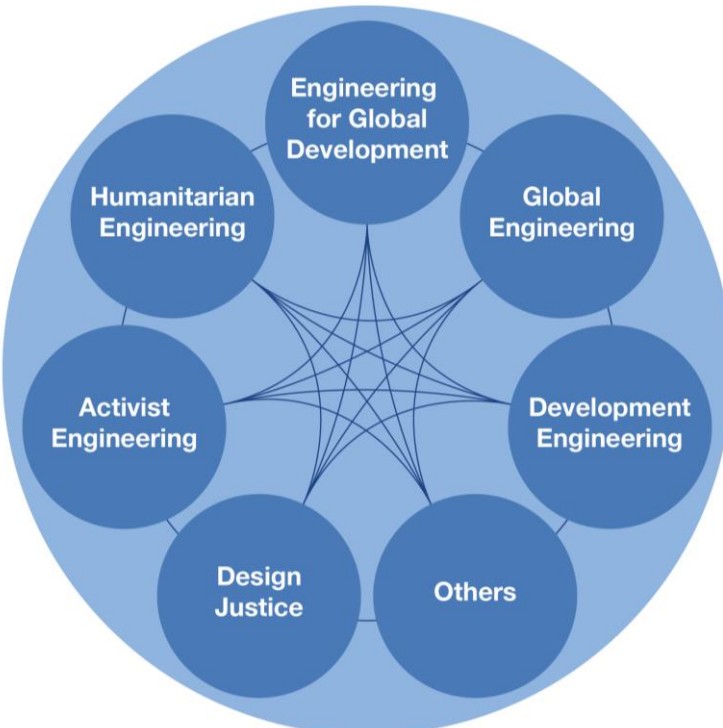

**Figure 6.** The interconnectedness of engineering areas related to sustainable development.

*4.2. Emerging & Recommended Cultural Shifts for the Field at Large*

The challenges that ESD efforts seek to address are wide-ranging and intertwined. Complex problems cannot, by definition, be solved by discrete solutions or processes. Based on the findings from the summit, ESD thought leaders are adopting and suggesting the following mindset and cultural shifts (i.e., reconstruction of concepts and definitions within engineering) to address challenges of this magnitude, urgency, and complexity:

- Establishing approaches to deal with complexity;
- Increasing representation;
- Rethinking the nature of engineering.

These cultural shifts are depicted in Figure 7 and discussed further below.

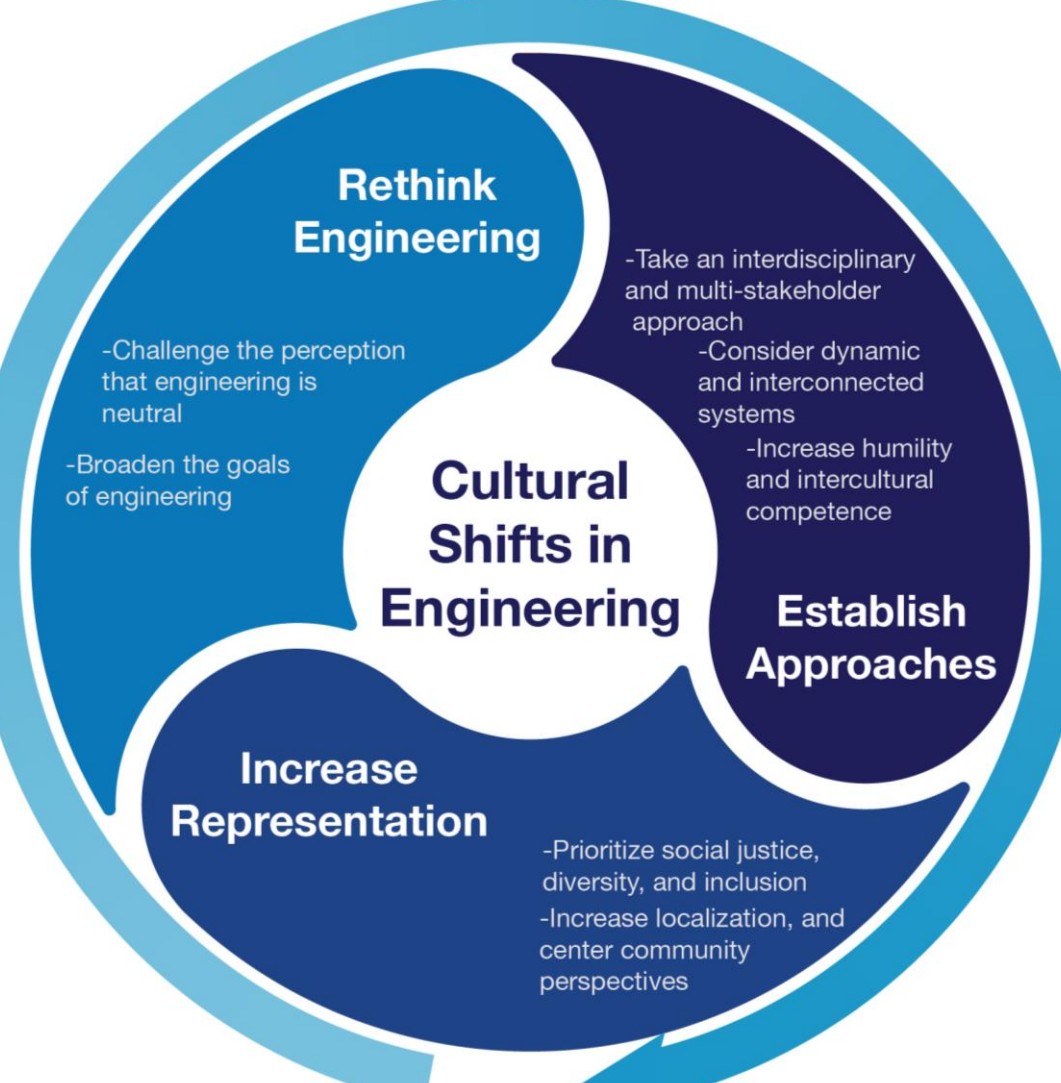

**Figure 7.** Emerging and recommended cultural shifts in engineering needed to address wicked problems, particularly in sustainable development.

4.2.1. Establishing Approaches to Deal with Complexity

Recommendation 1: Take an interdisciplinary and multi-stakeholder approach. Within ESD, there are many varying disciplinary standards of work [53] and differences in engineering-specific technical expertise and knowledge domains [7,54]. As participants identified, the most effective solutions will be those that (1) cross existing sector silos [55,56]; (2) bring industry, academia, NGOs, and the public sector together; (3) cross geographic and

socio-cultural boundaries and advance cross-cultural collaboration; and (4) engage with multiple perspectives, disciplines, and knowledges. Given the impact needed, multiple schools of thought and sectors are needed to address ESD-related challenges. The field must expand its awareness and acceptance of different perspectives and approaches. Notably, other investigations have identified that all fields of engineering (not just ESD) increasingly value sustainability and systems-engineering skills [6]. The engineering field is trained to prioritize technological solutions to challenges, even when other forms of innovation and knowledge may be better suited to addressing them. Our findings indicate that building relationships with fields outside of engineering, and taking their perspectives seriously, must become a regular practice within this work.

Recommendation 2: Consider dynamic and interconnected systems. Results from the summit suggest that engineers must develop and apply systems-thinking capabilities to define and address complex and multi-faceted problems. Ignoring the broader dimensions of systems beyond the technical (e.g., socio-economic, environmental, political, cultural, historical) and how they interconnect leads to failures and negative impacts within ESD and when addressing engineering problems more broadly [57]. COVID-19 forced the engineering field to reckon with the larger systems that drive our world. The pandemic contributed to, among countless other effects, profound disruption in healthcare, labor markets, education, supply chains, and policymaking processes. These interconnections also play out in less obvious and smaller-scale ways, for example, when introducing technology into an unfamiliar context without consideration for how social norms, culture, politics, or other factors might affect the level of impact. Both ESD challenges and potential solutions operate within dynamic and interconnected systems, and it will be critical to adopt socio-technical thinking [14], careful problem identification and scoping, and flexible approaches that collect and incorporate feedback quickly, and attempt to manage for primary as well as secondary and tertiary effects.

During the summit, the term 'systems' was discussed at length. In engineering, the term 'systems' can have a precise definition (e.g., related to the boundary drawn to model a mechanical, chemical, or thermal system), a wide-ranging definition (e.g., the broad socio-technical considerations surrounding a problem or solution), or a definition somewhere in between these two poles. Based on the findings from the summit, as well as recent literature, we recommend the use of 'systems' within engineering for sustainable development to consider the broader socio-technical considerations [14] as well declarations of its purpose, elements, and interconnections [58,59].

Recommendation 3: Increase humility and intercultural competence. Participants advocated for structural changes in engineering that increase funding and decision-making power to stakeholders and engineers in Low- and Middle-Income Countries. Participants also advocated for reducing the Eurocentric voices that dominate the field while embracing a sense of humility and developing intercultural competencies, which include open-mindedness, collaborative communication, and empathy [60,61]. Here, the term 'Eurocentric' is used to describe a focus on the values, experiences, and culture of Europeans and their descendants, i.e., 'the west' [62]. A sense of humility is particularly needed to break down the false notion of Eurocentric superiority. Terms like 'global competency', which is defined as the "knowledge, ability, and predisposition to work effectively with people who define problems differently than they do" [23], underpin a growing push for socio-technical competencies among engineers, conceived primarily as the ability to develop ethical engineering solutions attentive to the diverse cultures in which they are embedded [63]. The imperative to expand the breadth of skills engineers need also aligns with a recent Deloitte report that stressed the need to "prioritiz[e] enduring human capabilities" to ensure the long-term success of employees and organizations by developing fundamental skills such as communication and teamwork, as well as personal attributes such as empathy and humility [64]. More specifically, our findings advocate for engineers to adopt *epistemic humility*, which is the realization that one's knowledge is always incomplete and may require revision in light of new perspectives.

4.2.2. Increasing Representation

Recommendation 4: Prioritize social justice, diversity, and inclusion. Results from the summit explicitly connect social justice and sustainable development. Without social justice (i.e., providing people with fair opportunities and resources to live and thrive), we cannot achieve sustainable development or sustainability. Engineers often shy away from social justice because it challenges one to be very explicit about power, including power distribution that is hindered or facilitated by the technologies we design and build.

Engineering as a whole, particularly ESD efforts, needs a broad range of social identities [65], perspectives, backgrounds, skill sets, and knowledge domains to address complex engineering challenges holistically. Engineering institutions must work to cultivate inclusive and regenerative environments rather than ones that instill trauma in women, people of color, and other historically-excluded identities in engineering [66]. Training engineers to take an interdisciplinary and systems-based approach (Recommendations 1 and 2) may contribute to creating more inclusive environments, but additional action is needed. Many institutions have been working on improving diversity and equity in engineering for decades; aligning with and supporting existing societies, such as the National Society of Black Engineers, the Society of Hispanic Professional Engineers, the American Indian Science and Engineering Society (AISES), and the Federation of African Engineering Organizations, can help advance existing efforts. Other efforts may include outreach to pre-college students who are underrepresented in engineering. Outreach efforts should also emphasize the role of engineers in addressing sustainability and impact-driven problems. Researchers have found that framing engineering goals to include social justice considerations can help broaden the engineering field's diversity [29,67]. These efforts are needed to create an engineering field that reflects the compositional diversity of the world around it and supports its diverse members so that all can thrive in the engineering profession.

Recommendation 5: Increase localization and center community perspectives. For decades, western innovators have been developing solutions that they believe will be well-suited for low-resource environments in the 'developing world' and then handing them off (in a process that is sometimes referred to disparagingly as 'drag and drop') to local communities [68]. By and large, the result has been low uptake rates, technology that fails to fully consider the local context, and a limited engineering workforce in these environments. In some cases, these negative impacts have been devastating, resulting in detrimental health impacts (e.g., an arsenic crisis in Bangladesh due to the implementation of groundwater wells [69]), human rights violations (e.g., PlayPump [70]), and loss of millions of dollars in investments (e.g., One Laptop Per Child [71]). Instead, more significant effort should be dedicated to increasing localization, working with local partners to advance towards two-way skill-sharing, know-how transfer, capacity building, targeted support, co-construction of design solutions [72], and locally controlled long-term investments in infrastructure and services. Localization should embrace strengths-based approaches to recognize what expertise, skills, and insights local partners bring rather than a deficit-based approach. Towards this goal, there should be active advocacy for the reduced role of Eurocentric professionals and a reallocation of funding and decision-making power towards engineers and other partners based within the local setting. Many international ESD organizations aim to address this need by hiring regional directors who are experienced local leaders and experts in their country's context. Care is needed to ensure these efforts empower partners instead of tokenizing them [73]. The Design Justice Network (DJN) [74] provides ten relevant guiding principles, including using design to sustain, heal, and empower communities and seek liberation from exploitative and oppressive systems (DJN Principle 1), prioritizing the design's impact on the community over the intentions of the designer (DJN Principle 3), and working towards sustainable, community-led and -controlled outcomes (DJN Principle 8). Indeed, the field must continue to move away from a traditional 'save the poor' model towards interactions that ensure community empowerment [53].

### 4.2.3. Rethinking the Nature of Engineering

Recommendation 6: Challenge the perception that engineering is neutral. Participants highlighted how engineering has historically been framed as apolitical and divorced from historical and social contexts, phenomena that Cech calls the 'culture of disengagement' [75] and the 'ideology of depoliticization' [76]. Participants suggested that engineers broaden their definition of 'politics' to consider the affordances of power or exclusion in society, including their power to influence outcomes [77]. Our findings highlight the need for engineering to acknowledge the lack of neutrality—technology and sustainability goals and values are derived from individual and collective mindsets and worldviews [78].

Summit participants and scholars highlight the salient role of individual and collective identities and their relation to design and engineering outcomes; groups can be marginalized, harmed, and erased [51,79]. Engineers should be critical of their data sources since data generation and interpretation are not inherently neutral: algorithms, sampling protocols, categories, and measurement scales are all designed and debated [80]. Some resources exist to help engineers reflect and explore their work's social, environmental, economic, and peace implications [81]. For example, some industry leaders and scholars have invested in research and operationalizing robust ethics for advanced technology, such as AI and automation [82]. However, much work must be done to increase widespread awareness and consideration of values and biases among engineers, particularly those in ESD who aim to improve social equity. It has been and continues to be tempting to treat technology as a set of apolitical tools that impact the world only through the intentions of their users and to believe that technological improvements are always positive. Instead, engineers need to adopt a historically grounded perspective, as many of the world's most pressing challenges can be directly traced to past technological changes. More significant initial consideration of what that technology might have been used for could have prevented these challenges from arising.

Recommendation 7: Broaden the goals of engineering. Following the lead of organizations, such as the Tech Stewardship Network and Design Justice Network, the field should reconsider what engineering is, what it includes, and who and what it is for [83]. Historically, there has been a perception that engineering solely aims to produce systems and products, often with primary goals of technical performance, efficiency, and profitability [84]. There is an opportunity for engineering to address other metrics and goals, such as poverty reduction, climate, sustainability, and social justice. However, it must address how it has contributed to the challenges it seeks to solve (e.g., the relationship between fossil fuel-burning technology and climate change; the creation of technology that makes warfare more deadly). Training in critical thinking regarding the social, environmental, and economic impact of engineering products along their life cycles will be essential.

One way to engage engineers in broader societal issues is through policy. Most engineering associations and companies engage closely with government stakeholders, providing technical perspectives to conversations about regulations and policy changes. In this way, engineers have a great opportunity to participate in the political arena not just as individual civilians, but as professionals, offering much-needed technical perspectives. Coalitions, such as the Engaging Scientists & Engineers in Policy (ESEP), the IEEE European Public Policy Committee (EPPC), and the African Technology Policy Studies Network (ATPS), aim to bring engineering expertise into policy and governance discussions. There are also many opportunities for the political participation of engineers associated with political advocacy groups, such as those defending digital rights (e.g., Center for Digital Democracy, Accountable Tech, Fight for the Future), promoting water, energy, and technology access (e.g., International Energy Agency, Clean Water Action, First Nations Technology Council) and climate action (e.g., Climate Action Network, Natural Resources Defense Council). Engineers can also be valuable contributors to "think tanks" and other research organizations that conduct policy-driven research. Advocacy and research groups such as these can offer engineers opportunities to learn from, inform, and drive sound policymaking that reduces inequalities. Involvement in policy also helps shape engineers

that are more informed of the broader context of their work and the impact policy has on shaping engineering priorities. For example, a growing number of clean energy goals and mandates in the U.S. and globally will require engineering advances in battery storage technology, developing a more sustainable lifecycle for renewables, and grid hardening to improve resilience to climate hazards such as heat and flooding. Awareness of and attention to these large-scale policy drivers can inform more impactful engineering.

### 4.2.4. Cross-Cutting Themes

While these cultural and mindset shifts are essential on their own, they are also mutually reinforcing, and gains in one of these areas can lead to gains in another. For example, improved skills in intercultural competence and cultural humility (Section 4.2.1) can help to foster inclusion and enable diversity (Section 4.2.2) [85–88]. Taking an interdisciplinary and multi-stakeholder approach (Section 4.2.1) that incorporates socio-technical perspectives [89], such as from Science & Technology Studies, can provide a new lens for engineers to consider the perspective that technology and engineering are not neutral (Section 4.2.3). Finally, reframing and broadening the goals of engineering to include social justice and equity (Section 4.2.3) can additionally serve to broaden the diversity of the engineering field (Section 4.2.2), attracting under-represented students who value helping others over maximizing profits [67].

### 4.3. Incentives, Metrics, and Decision-Making Frameworks

These big-picture shifts in thinking must be accompanied by processes for implementing them, including decision-making frameworks and regulatory frameworks. At the highest levels, companies, academic institutions, NGOs, and political actors must find a way to prioritize ethics and sustainability. As organizations make this prioritization, they will need to explore what ethics and sustainability mean in practice. Actionable steps may include deciding what indicators should be used for tracking progress, how to weigh trade-offs, how to manage competing priorities, and what levers stakeholders can use to incentivize adherence and accountability. Regulations are critical in motivating organizations to make more sustainable and ethical decisions; those with experience and training in ESD have an essential role in advocating, operationalizing, and advancing regulatory frameworks.

Economic incentives will be particularly vital; if companies see ESD as a pure cost rather than an opportunity, they are unlikely to prioritize it. Thus, the sector as a whole must find a way to align progress towards ESD goals with companies' bottom lines. When companies find ways to do well by doing good, it will be critical to highlight success stories and create continued positive social pressure on others to learn from and replicate their approaches. However, it will be critical to incorporate measures to ensure that companies do not fall into 'impact washing', 'ethics washing', or 'greenwashing', in which the evidence of their reported impact is insufficient. An increasingly important evaluation and reporting tool is the Environmental, Social, and Governance (ESG) framework adopted by investors and shareholders who want to improve company sustainability. In addition to metrics related to the Sustainable Development Goals, which many global development organizations employ, organizations could implement the ESG framework to align with trends in the private sector. The ESG efforts of an organization have also been identified as crucial for recruiting and retaining engineers. A recent survey found that an employer's ESG activity and rating influence the decision-making process of 61% of engineers when deciding whether to remain at their company [90].

Importantly, once an organization commits to improving its social and environmental impact, engineering organizations will need accompanying key performance indicators (KPIs), processes for measuring and reporting impact (including codified metrics and reliable sources of data), and access to training on standards, assessments, and other relevant tools to make these high-level commitments tangible. These tools must apply to big-picture decision making and the day-to-day actions of individual engineers, which have

been referred to as micro- and macro-ethics in engineering [91]. While many frameworks and metrics exist for ESD efforts, guidance and regulation remain lacking. Moreover, as previously mentioned, there exists a gap in data expectations: both in terms of availability and reliability of data to make equitable and dependable design decisions.

Within ESG, tools for understanding environmental impacts, such as life cycle assessment with its ISO standard [92], are more established than in the less mature field of social impact [93]. Some assessment tools exist for community-level impact assessment, e.g., social impact assessment [94] and social return on investment (SROI) [95], but their application and use are varied. For the social impact of products, methods such as social life cycle assessment [96] have emerged. However, methods for implementing them have not been standardized [97]. Selecting what social impact metrics to use for measuring social impact has remained a challenge. Scholars have characterized eleven categories of social impacts, e.g., population change, gender, and human rights [98], but standardized metrics for these categories remain lacking. Within product development, scholars have proposed social performance indicators related to employment, health, and standard of living, among others [99,100], and there have been recent advancements in the modeling and prediction of social impacts of engineered solutions [101,102]. However, ample opportunity remains to advance and operationalize socially driven engineering impacts and goals.

*4.4. Engineering Education*

We advocate for a different kind of engineering education that goes beyond broader skills and interdisciplinarity to bridge the gap between engineering education and sustainable development. Content and pedagogies must be rethought since they have historically positioned the economic advancement of a few at the expense of many [103]. For example, teaching thermodynamics with the internal combustion engine has served the automotive industry's needs. However, as we move to decarbonize our modes of transportation (and perhaps eventually phase out of internal combustion engines altogether), technical skills and their applications must be rethought. Our findings have scratched the surface of what it could look like to rethink engineering education, and we recommend future work considers a more drastic reframing of engineering education.

Based on the existing limitations in engineering education, our findings suggest various curricular, co-curricular, and extra-curricular training opportunities for engineering students to develop socio-technical skill sets and robust critical thinking. There is a need to expose university students and early-career engineers to various possible career paths in and beyond ESD. We recommend a reframing away from ESD-specific career paths toward a much broader philosophy that ESD principles should be embedded within all career paths. Every engineer should constantly examine and consider their work's favorable and unfavorable economic, social, environmental, local, and global consequences. As previously discussed, younger generations increasingly seek careers that reflect broader social and environmental values. These ESD values need to come to the forefront within all engineering careers. By equipping the next generation of engineers to come to their work with the relevant tools and mindsets, we would anticipate ESD philosophies to permeate throughout the profession.

To promote the inclusion of ESD training and principles, institutions could focus on incentive structures, including recognition and certification of socio-technical skills, as well as incorporation into evaluation mechanisms used for promotion and tenure for its faculty. Additionally, to increase the diversity of the graduating workforce, there should be increased inclusion and equity efforts among universities, primarily in retaining historically excluded individuals in engineering. Some universities have made critical declarations and commitments to equity-centered engineering [104], but these efforts require continuous resources and funding to remain sustainable.

Further identification and definition of shared goals and language between academia and industry would improve synergy and support the development of practice-based learning approaches that model critical-thinking and problem-solving skills. Engineers

want efficient tools to make sustainable decisions, and executives want to publish accurate statements, but tools to evaluate impact are lacking or complex (and often, third-party experts perform this work for engineering firms); there are undoubtedly ample career opportunities for engineering graduates to advance and perform impact assessments (e.g., life cycle assessment (LCA), stakeholder analysis, systems thinking) and research opportunities for academic institutions to support engineering decision frameworks.

### 4.4.1. Curricular Education

Integrating sustainable development practices and the principles identified in this study are not simply additional material to be included in existing curriculum. Instead, teaching needs to be transformed to encourage and equip students to be more inclusive and more open to social and environmental responsibility. One example of such transformation is from UniLaSalle, a polytechnic university in France, where a 'Living Lab' was developed to provide students with sustainable development activities that are situated in real-world contexts [105].

Within traditional engineering classrooms on campus, there is a perception that training socio-technical competencies and systems thinking is complex and time-intensive. However, there are many resources for incorporating socio-technical considerations into design prompts, classroom exercises, and case studies used in technical classrooms [59,106–108]. Additional frameworks, such as Engineering for One Planet, provide a menu of sustainability learning outcomes. Project-based courses (e.g., introductory design, capstone design) are great opportunities to provide ESD training through hands-on examples [109]. Additionally, many institutions offer minors, certificates, and coursework focused on ESD [12] to integrate skills and theories from the social sciences and other disciplines, allowing students to integrate relevant training without significant time spent identifying and completing courses outside of their traditional engineering program. During the summit, participants highlighted that within the U.S., much work can be done to expand ABET requirements regarding broader social, political, and environmental contexts within engineering education. Moreover, we recommend further integrating International Engineering Alliance (IEA) Graduate Attributes and Professional Competencies, which were updated in 2021 to emphasize diversity and inclusion, sustainable development, and UN SDGs [3].

### 4.4.2. Co-Curricular and Extra-Curricular

Co-curricular activities can provide students with experiences outside the traditional classroom material [110]. For example, the Engineers Without Borders (EWB) Challenge in Australia and New Zealand allows over 10,000 undergraduate students to engage in a team-based humanitarian engineering project. The EWB Australia Capstone research program annually supports hundreds of students undertaking humanitarian engineering capstone projects. The EWB Australia Design Summit program has offered almost two thousand students immersive humanitarian engineering design experiences in developing countries. Similarly, the Siemens Design Challenge hosted by Engineering for Change to develop technology-based solutions for challenges related to the Sustainable Development Goals drew over 23,000 people from 184 countries and generated more than 220 ideas at the height of the COVID-19 pandemic. Collaborations between universities and engineering organizations can also be expanded to offer more practical work experiences (e.g., via co-ops or internships) that model what undergraduates will experience in their careers while also providing upskilling and reskilling training opportunities for employees. Participants in the summit noted the current tension that many engineering opportunities in global development are under or unpaid, suggesting that universities help subsidize these experiences through scholarships or fellowships.

### 4.4.3. Post-Educational

There is an opportunity to extend ESD education to engineers who have already completed their university education. A broader reach of ESD training opportunities, such

as the Engineering for Change Fellowship and the Tech Stewardship Practice Program, can allow more engineers to gain valuable exposure and skills in this field. Moreover, with the expansion of Massive Open Online Courses (MOOCs), there is an opportunity to reach engineers with busy schedules via online educational platforms such as Coursera and edX.

*4.5. Limitations*

This research only investigated the views and perspectives of people with connections to ASME and EGD, with a minimal exploration of how other perspectives in engineering (e.g., representatives from engineering firms that do not engage in or support sustainability initiatives) may contradict these views. Another limitation may stem from the self-selection biases of those who were motivated enough by these issues to participate in the summit. Future work could explore the generalizability of the trends presented in this work and verify these shifts' impact on achieving sustainability outcomes.

**5. Conclusions**

This study explored emerging trends in advancing engineering to support the UN SDGs. We applied an adapted Delphi methodology to bring together over 50 leaders in industry and academia with a variety of experiences in ESD. We conducted surveys, facilitated focus-group discussions, verified our findings through semi-structured interviews, and developed recommendations through iterative thematic analysis.

Engineering for Sustainable Development is no longer a niche field; it is now a robust community of practitioners and academics. Its scope has expanded thanks to an increased demand for impactful careers and sustainability-focused work, as well as a growing recognition of the connections between disciplines and projects. There is increasing awareness of the mindsets, training, and tools engineers need to address global sustainability challenges in an increasingly globalized workforce. Among these mindsets are taking an interdisciplinary and multi-stakeholder approach, considering dynamic and interconnected systems, prioritizing diversity and inclusion, challenging the perception that engineering is neutral, increasing humility, localization, and centering community perspectives, and broadening the goals of engineering. Cross-cultural collaboration and leadership are increasingly desired skills for all engineering professionals. ESD and its related fields and programs are well positioned to provide education and training in these areas. With a growing demand for impactful and sustainable work and increased recognition of the interdisciplinary nature of sustainability challenges, engineers must embrace a new set of values and redefine their role. By prioritizing diversity, inclusion, cross-cultural collaboration, and community perspectives, engineers can become leaders in addressing global sustainability challenges and creating a more just and equitable world.

**Author Contributions:** Conceptualization, I.A.; methodology, I.A., G.B., J.L. (Jason Lajoie), S.D. and D.K.; formal analysis, J.L. (Jason Lajoie), G.B., C.M. (Christopher Mabey), E.P. and M.S.K.; investigation, J.L. (Jason Lajoie), G.B., C.M. (Christopher Mattson), A.W., E.T., L.M., E.P., M.S.K., P.S. and J.V.; data curation, J.L. (Jason Lajoie), G.B., C.M. (Christopher Mabey), E.P., M.S.K., P.S. and J.V.; writing—original draft preparation, G.B., J.L. (Jason Lajoie), E.S., D.K., S.D., C.M. (Christopher Mattson), J.V. and P.S.; writing—review and editing, E.T., J.L. (Jason Lajoie), G.B., C.M. (Christopher Mabey), E.P., M.S.K., J.A.-B., C.M. (Christopher Mattson), A.J.-W., J.V., P.S., J.L. (Juan Lucena), and D.K.; visualization, C.M. (Christopher Mabey), J.L. (Jason Lajoie), and G.B.; project administration, I.A.; funding acquisition, I.A. All authors have read and agreed to the published version of the manuscript.

**Funding:** This research was supported by the American Society of Mechanical Engineers (ASME).

**Institutional Review Board Statement:** This study was conducted in accordance with the Declaration of Helsinki, and was approved by the Engineering Global Development Committee of ASME, the associated ASME Program and Philanthropy business units and Engineering for Change research management committee.

**Informed Consent Statement:** Informed consent was obtained from all participants involved in the study.

**Data Availability Statement:** Data can be made available upon request.

**Acknowledgments:** We express our gratitude to the ASME EGD and Engineering for Change staff and leadership for their technical support and additional contributions to the summit.

**Conflicts of Interest:** The authors declare no conflict of interest.

## Appendix A. Surveys

Survey 1: Industry participants

1. What is your name? (First & Last)
2. What is your email?
3. Which of the following industries best represents your profession?

    a. Agriculture/Other rural sectors
    b. Metal production
    c. Chemical
    d. Commerce
    e. Construction
    f. Education
    g. Financial services/Professional services
    h. Food/Drink/Tobacco
    i. Forestry/Wood/Pulp and paper
    j. Health services
    k. Hotels/Tourism/Catering
    l. Mining
    m. Media/Culture
    n. Oil & gas/Oil refining
    o. Telecommunications
    p. Public service
    q. Shipping/Ports/Fisheries/Inland waterways
    r. Textiles/Clothing/Leather/Footwear
    s. Transportation
    t. Utilities
    u. Other

4. What's your role and organization?
5. What's the size of your organization?

    a. Microenterprise: 1 to 9 employees
    b. Small enterprise: 10 to 49 employees
    c. Medium-sized enterprise: 50 to 249 employees
    d. Large enterprise: 250 employees or more

6. What's the size of your current team? Type in the number of people on your specific team within your organization.
7. Approximately what percentage of your team are engineers?
8. What is the total projected engineering hires for next three years (including full-time hires, interns, co-op, temporary hires, and contractors)?

    a. None
    b. 1–5
    c. 6–20
    d. 21–100
    e. 101+
    f. I don't know

9. What are the engineering disciplines for your projected hires?

    a. Mechanical engineering
    b. Electrical engineering
    c. Chemical engineering

    d.    Civil engineering

    e.    Industrial engineering

    f.    Systems engineering

    g.    Computer engineering

    h.    Other

10. How much do you agree/disagree with the following statement? "Engineers trained in sustainable development are valuable to our organization"

    a.    Strongly disagree

    b.    Disagree

    c.    Neutral

    d.    Agree

    e.    Strongly agree

11. Why is your organization interested in hiring engineers trained in sustainable development? (Choose as many as you like)

    a.    Company's focus in sustainable business practices

    b.    Growing public interest in sustainable business practices

    c.    Investors interest in companies addressing the Sustainable Development Goals

    d.    Company's focus on innovation

    e.    Other

12. What are the top 3 skills that you most value in your ideal engineering candidates with experience in sustainable development? (Choose up to 3 responses)

    a.    Systems thinking

    b.    Design

    c.    Problem solving

    d.    Innovation and creativity

    e.    Verbal and written communication

    f.    Team work

    g.    Cross-cultural collaboration

    h.    Time management and project management

    i.    Leadership

    j.    Work ethics

    k.    Knowledge about emerging technologies

    l.    Knowledge about sustainable development

    m.    Diverse points of view

    n.    Other

13. What skills would your organization like to see more of in your engineering workforce? (Choose as many as you like)

    a.    Systems thinking

    b.    Design

    c.    Problem solving

    d.    Innovation and creativity

    e.    Verbal and written communication

    f.    Team work

    g.    Cross-cultural collaboration

    h.    Time management and project management

    i.    Leadership

    j.    Work ethics

    k.    Knowledge about emerging technologies

    l.    Knowledge about sustainable development

    m.    Diverse points of view

    n.    Other

14. How much do you agree/disagree with the following statement: "When considering how the practice of engineering will change in response to the Sustainable Development Goals, I feel my organization is well prepared".
    a. Strongly disagree
    b. Disagree
    c. Neutral
    d. Agree
    e. Strongly agree
15. What are your key reasons for your answer to the previous question?
16. Whithin the next three to five years, opportunities to practice sustainable development within our organization are likely to . . .
    a. Strongly decline
    b. Decline
    c. Stay the same
    d. Increase
    e. Strongly increase
17. What are some career paths for engineers to practice sustainable development in your organization? (present or future)
    a. Analyses of project alternatives
    b. Feasibility studies
    c. Environmental impact studies
    d. Assistance in project planning, approval, and financing
    e. Design and development of sustainable systems, processes and products
    f. Design and development of sustainable construction plans
    g. Project management
    h. Sustainable construction supervision and testing
    i. Social innovation/entrepreneurship
    j. Management consulting
    k. Environmental monitoring
    l. Decommissioning of facilities
    m. Restoration of sites for other uses
    n. Resource management
    o. Design and development of innovation programs
    p. Other
18. What factors are enabling these pathways for engineers to practice sustainable development (now or in the future)?
    a. Partnerships
    b. Mentorship
    c. Training and education
    d. Networks and communities
    e. Collaboration and acceleration platforms
    f. Funding
    g. Other
19. Optional: Are there any special skills/knowledge that your organization requires for these pathways?
20. Do you have any comments you'd like to add before submitting the survey?

    Survey 2: Academic participants

1. What is your name? (First & Last)
2. What university are you affiliated with?
3. What is your email?
4. What is your primary discipline? (select all that apply)
    a. Bioengineering/Biomedical engineering

    b.     Civil engineering
    c.     Environmental/Chemical engineering
    d.     Electrical/Computer engineering
    e.     Engineering education
    f.     Global engineering/Humanitarian engineering
    g.     Industrial engineering
    h.     Mechanical engineering
    i.     Software engineering/HCI
    j.     Systems engineering
    k.     Urban planning/Architecture
    l.     Other

5.    Which of the following best describes your title?

    a.     Student
    b.     Post-doctoral scholar
    c.     Instructor
    d.     Assistant professor
    e.     Associate professor
    f.     Professor
    g.     Associate dean
    h.     Dean
    i.     Other

6.    How much do you agree/disagree with the following statement: "Training engineers to work in EGD*-related careers is core to my teaching philosophy" (*EGD = engineering for global development)

    a.     Strongly disagree
    b.     Disagree
    c.     Neutral
    d.     Agree
    e.     Strongly agree

7.    What opportunities do you specifically lead/mentor to provide EGD-related training for students? (Choose as many as you like)

    a.     EGD-focused degree/certificate programs
    b.     Research
    c.     Classes focused on sustainable development, engineering ethics, etc.
    d.     Capstone projects within EGD-contexts
    e.     Clubs/organizations (e.g., EWB)
    f.     Internship/study-abroad experiential programs
    g.     Other

8.    What are the top 3 skills that you train your students preparing for EGD-related careers? (Choose up to 3 responses)

    a.     Systems thinking
    b.     Design
    c.     Problem solving
    d.     Innovation and creativity
    e.     Verbal and written communication
    f.     Team work
    g.     Cross-cultural collaboration
    h.     Time management and project management
    i.     Leadership
    j.     Work ethics
    k.     Knowledge about emerging technologies
    l.     Knowledge about sustainable development

　　　m.　Diverse points of view
　　　n.　Other

9.　Tell us more: In your experience, what types of learning opportunities provide students with the best training/preparation for EGD-related careers? (open response)
10.　Approximately what percentage of your time/effort goes towards EGD-related work (compared to more traditional disciplines in engineering)? (response: x%)
11.　In an ideal world, what would you want this percentage to be? (response: x%)
12.　List 1–3 factors that are barriers for working on more EGD-related projects. (open response)
13.　Within the next three to five years, opportunities for students to pursue EGD-related training at your institution are likely to . . .

　　　a.　Strongly decline
　　　b.　Decline
　　　c.　Stay the same
　　　d.　Increase
　　　e.　Strongly increase

14.　What are some EGD-related career paths for engineers that you mentor students towards now or in the future? (Choose as many as you like)

　　　a.　Academic careers (grad school, faculty career paths)
　　　b.　Analyses of project alternatives
　　　c.　Feasibility studies
　　　d.　Environmental impact studies
　　　e.　Assistance in project planning, approval, and financing
　　　f.　Design and development of sustainable systems, processes and products
　　　g.　Design and development of sustainable construction plans
　　　h.　Project management
　　　i.　Sustainable construction supervision and testing
　　　j.　Social innovation/entrepreneurship
　　　k.　Management consulting
　　　l.　Environmental monitoring
　　　m.　Decommissioning of facilities
　　　n.　Restoration of sites for other uses
　　　o.　Resource management
　　　p.　Research and development (R&D)
　　　q.　Measuring progress for sustainable development
　　　r.　Other

15.　What are you finding to be enablers for your students to pursue EGD-related career paths?
16.　Do you have any comments you'd like to add before submitting the survey?

**Appendix B. Example of the Content Developed for the Focus-Group Sessions Using a Virtual Whiteboard Platform**

　　　Virtual whiteboards were designed for all session topics as a way to ground discussion and engage participants. This screenshot below presents an example from Session 2: Topic 2.2. Two comments are included on the whiteboard for participants to access; one reads: "Goal: Identify and describe existing and potential barriers to thriving EGD careers, as well as existing and potential enablers". The other comment reads: "Instructions: List factors under the appropriate column based on whether it will enable engineers to practice sustainable development or constrain their ability to do so. Describe the career pathways in your organization that are most likely to be impacted by each influence".

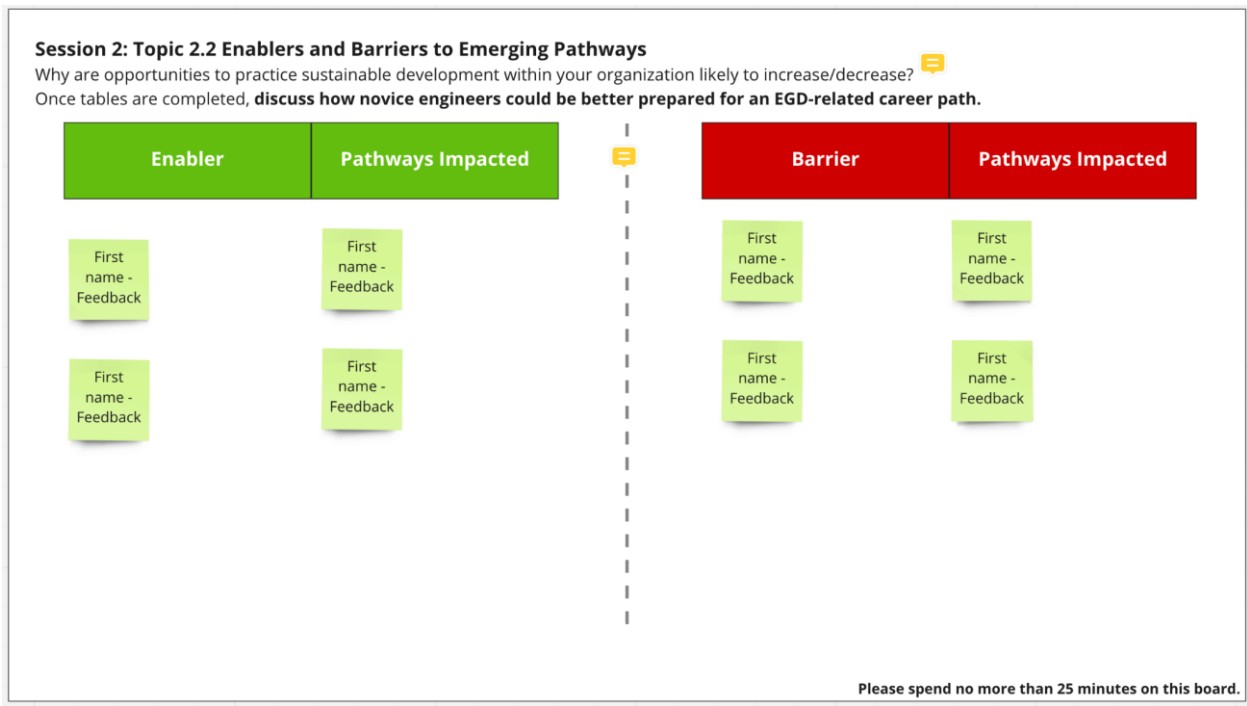

**Figure A1.** Example of virtual whiteboard used during Focus Group Sessions.

### Appendix C. Excerpt from One of the Minutes Drafted from a Focus-Group Session and Example of Virtual Sticky Notes Collected during One Session Topic in One Group

Excerpt from minutes drafted during a focus-group session for Topic 2.1. All names have been removed and presented as "facilitator" or "participant" and identifying information (e.g., organization names) have been redacted.

[Facilitator] briefly reviews survey results; instructs folks to provide examples and place on the graph

[Participant]: sees a growth in fund managing in local context; need engineers that understand commercializing paths; emergence of funds managers as engineers; about 5 years ago it used to be traditional financing people

[Participant]: engineering and human rights; partner with human rights center; exs. Finding graves of people that have disappeared; assistive devices, internet access

[Participant]: people that don't have access to engineering benefits; try to get students to think about low cost alternatives; collaborative market; accessible off-grid industries partners/internships

[Participant]: assistive technologies; STEM education for people who are visually impaired in [redacted]; STEM accessibility; Technology center locally; sensor/wearable systems

[Facilitator] provoked room to think about past students and where they have ended up

[Participant]: working for NGO or nonprofit; has interest to see more pathways; find the market for students; students with these aspirations find things in development like [redacted], see students grow into management roles; interested to learn from industry

[Participant]: more interest in agencies to be bridge between entrepreneurs that are building technologies to agencies that regulate these technologies; no one agency in charge of pathway; not appropriate to spend funding on one-of regulatory processes; how to stop the default being importing technologies; engineers could be a strong contributors to these conversations

[Participant]: [Redacted]'s work and book helps us think about how engineers engage in CSR

[Participant]: maybe a requirement for students; that they must engage in CSR; might help them prioritize it in their studies

[Participant]: pathways are very near and dear to my heart; I think this will be the most exciting decade of engineering; I am jealous of people that are starting their career; engineering community will be involved with this like space exploration; there are things that we can bring back from space to help in energy, resources; maybe think about artificial limbs in zero G; they grow faster out there; bridge academia and industry to take up these things that are going on; young people seem to be kinder and into entrepreneurship; help younger people follow their aspirations

[Participant]: as a [redacted]; the pathways are completely different between global north or global south; those opportunities were not available in [redacted], particularly in architecture

[Facilitator]: survey results about enablers and obstacles/barriers

[Participant]: if you have a cause; you will find investors; you have to communicate and articulate the value of what is being done under these agencies; there are investors out there; in Silicon Valley people make a lot of money very quickly, they want to help others have the same opportunities; there are investors out there if we are able to articulate the cause

[Participant]: barriers; technology solely; need the integration of social science, ethics, in a systems way; we hear from engineering faculty that the work is "not technical enough"; want us to save a community, like a white savior mentality that we try to avoid; or they want something really tech savvy

[Participant]: how do we address these issues?

[Participant]: the systems approach in engineering; here as well; ABET; engineering educators; funders; alignment with the message; technology stewards; [redacted]; we are talking to the choir, how do we bring in the ones that aren't involved; how do bring in the people that don't think it's technical enough, entrepreneurial enough, etc.

[Participant]: I have found this too; I wonder if they are not the people to convert; decision makers higher up; maybe there is advantage of having a choir;

[Participant]: balance between experiential learning and ethical consideration; engineering do a lot of experiential learning, internships; how do we manage the risk on the communities that we are trying to serve and the learning opportunity of our students? How do we provide some guidelines for the balance?

[Participant]: champions of this movement; say they have these ah-ha moments in these international spaces; I would expect that they happened in ways that we wouldn't support now; less impactful

[Participant]: lots to say about ah-ha moments; how do we learn about these moments and try to replicate (maybe) them in our students

[Participant]: these people that run these organizations don't have experience in innovation; acknowledge that we have a lot of learning to do as a whole

[Participant]: we don't have a lot of mentors for students; we need more role models in academia; maybe this group can help; we as a community can do together; very interested in pathways but they don't see it

[Participant]: mentorship is really important; would being in a program that is not technical enough be a barrier for them? How do we think with them about these decisions

[Participant]: shared a virtual internship [redacted]

[Participant]: Promotion and Tenure policies; as non-academic it almost always comes up; PTIE; coalition about P&T; what would we want to change? Is it about entrepreneurship or innovation? Policies and culture changes; comes up continuously

[Participant]: [Redacted] is making some of these changes in this direction

[Participant]: Impact Investment—Increased number of analysts. The increased number of hardware solutions—engineers to work on this realm.

[Participant]: Engineering and human rights (working on this minor)—Use ArcGIS to find graves of people that have disappeared

[Participant]: Energy access—provide affordable alternatives—track to move into that market—a more collaborative market in that way. Connect off-grid industries with universities.

[Participant]: work with some organizations with an emphasis in STEM education—STEM accessibility.

[Participant]: Students finding opportunities in NGO from entry level to grow inside the organization.—interested to learn from the industry

[Participant]: be a bridge between the entrepreneurs of the health industry with agencies that work on regulations on the health environment.—Engineers play a huge role in these facilitations.

[Participant]: Engagement in CSR initiatives

[Participant]: Will the pathway for northern regions be different for southern regions?

[Participant]: this decade will be the more exciting to be an engineer, short start-ups

[Participant]: Experiential learning—How do we manage/balance the risk of these experiential learnings? Balance the trade-offs.

[Participant]: Student learning—Impact balance maybe could look different in the academic than from a personal perspective

[Participant]: multidisciplinary training—engage faculty in different roles of the EGD path. Some of the people that advocate these do not have that much in-hand experience.

[Participant]: No mentors or career models for students that will like to follow these career paths right now.

[Participant]: 10 year incentive—entrepreneur pathway to decrease the barriers in EGD paths. Cultural shift in organizations are needed to increase these opportunities

Example from virtual sticky note data, taken from one group's responses to the following prompt: "Make a sticky note for an emerging career pathway in your organization (or ones you know about elsewhere)"

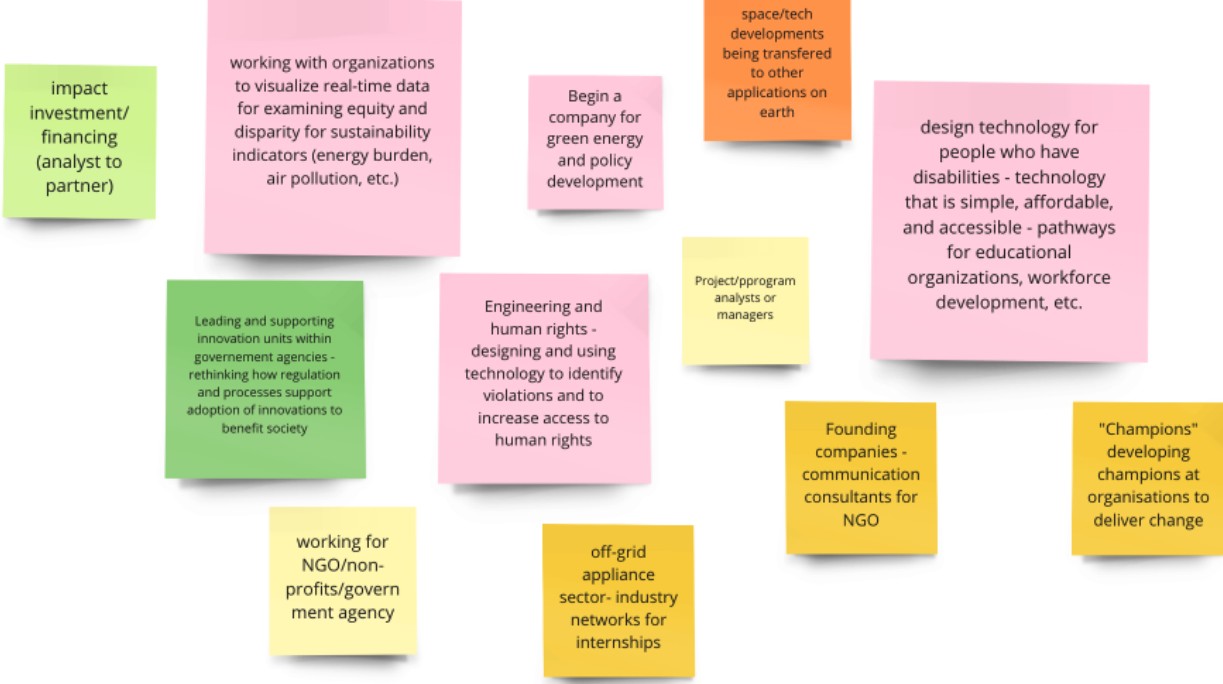

**Figure A2.** Example of data collected from virtual whiteboard 'sticky notes'.

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
