# Peer review of "Advancing Sustainable Development: Emerging Factors and Futures for the Engineering Field"

_sustainability, doi:10.3390/su15107869_

Round 1

Reviewer 1 Report

This study examines the emerging trends in the field of Engineering in the light of the UN SDGs. The topic is very relevant to the current needs and requirements of the field and the paper has been organized properly accordingly. The academic tone of the paper is also evident. This type of research is extremely needed in our days as the engineering profession is witnessing  over-saturation and is sometimes stagnant in some countries and surely there is a need for original reforms to take place. It fills a great gap between the literature and the practice of engineering. I only have one point to be considered by the authors that is related to the filed of engineering itself. Since the field is very diverse, can the others be more specific about which fields in Engineering are more likely to be more easily improved and which are likely to trigger resistance. I believe this is very important. This should be addressed in the literature review and the recommendations sections in more elaboration. Reforms for instance in electrical engineering could be, more or less, more flexibly implemented than structural engineering, I am assuming, and so on. Otherwise, I believe the paper has comprehensively and adequately tackled all the elements of the study adequately. The reference list is comprehensive and inclusive. The gap has been addressed and covered properly. 

Author Response

We (the authors) appreciate the time and effort you have made to review our manuscript. We have included your comments (bolded) and our response below. We believe that your suggestions have strengthened our manuscript.

This study examines the emerging trends in the field of Engineering in the light of the UN SDGs. The topic is very relevant to the current needs and requirements of the field and the paper has been organized properly accordingly. The academic tone of the paper is also evident. This type of research is extremely needed in our days as the engineering profession is witnessing  over-saturation and is sometimes stagnant in some countries and surely there is a need for original reforms to take place. It fills a great gap between the literature and the practice of engineering. I only have one point to be considered by the authors that is related to the filed of engineering itself. Since the field is very diverse, can the others be more specific about which fields in Engineering are more likely to be more easily improved and which are likely to trigger resistance. I believe this is very important. This should be addressed in the literature review and the recommendations sections in more elaboration. Reforms for instance in electrical engineering could be, more or less, more flexibly implemented than structural engineering, I am assuming, and so on. Otherwise, I believe the paper has comprehensively and adequately tackled all the elements of the study adequately. The reference list is comprehensive and inclusive. The gap has been addressed and covered properly.

Thank you for your review and we appreciate these comments. We agree that engineering is an extremely diverse field and that we previously ill-defined “engineering” in our original submission. To address this, we have included the following additional clarifying paragraph in the introduction:

“In recognition of these global challenges, efforts across various sectors and industries in the engineering field have sought to advance sustainability. The International Engineering Alliance defines engineering as a practice, body of knowledge, and set of techniques that can be purposefully applied to produce solutions of which the performance and impacts are predicted to the greatest extent possible (IEA, 2021). Many engineering organizations are moving away from disciplinary and siloed definitions, for example, the American Society of Mechanical Engineers (ASME) defines their mission as one that “promotes the art, science & practice of multidisciplinary engineering and allied sciences around the globe” (ASME, 2023). Engineering practice and knowledge can be applied to many different fields, including design, policy, business, and economics (MacDonald et al., 2022). In this manuscript, we employ these broad definitions of engineering—focusing on the large global field of technical applied sciences (ASME & Autodesk, 2022).”

Reviewer 2 Report

This is a good article that explores different points of view. I would recommend some articles that present the case of an engineering school in France and confirm some of your analysis and conclusions:

- https://www.tandfonline.com/doi/abs/10.1080/03043797.2018.1501348

- https://hal.science/hal-03251079/file/Presentation%20SEFI%20colloque%20-%20final.pdf (presentation of the study and of all the authors' studies on education for sustainable development in engineering schools).

Best regards and good luck with your research.

Author Response

We (the authors) appreciate the time and effort you have made to review our manuscript. We have included your comments (bolded) and our response below. We believe that your suggestions have strengthened our manuscript.

This is a good article that explores different points of view. I would recommend some articles that present the case of an engineering school in France and confirm some of your analysis and conclusions:

- https://www.tandfonline.com/doi/abs/10.1080/03043797.2018.1501348

- https://hal.science/hal-03251079/file/Presentation%20SEFI%20colloque%20-%20final.pdf (presentation of the study and of all the authors' studies on education for sustainable development in engineering schools).

Best regards and good luck with your research.

Thank you for your suggestion, we agree that including this case of an engineering school in France will strengthen our discussion of engineering education models. We particularly think it showcases and interesting model of engineering education, in this case “Living Labs”. We have included this in our discussion (Beginning of Section 4.4.1: Curricular Education).

“Integrating sustainable development practices and the principles identified in this study are not simply additional material to be included in existing curriculum. Instead, teaching needs to be transformed to encourage and equip students to be more inclusive and more open to social and environmental responsibility. One example of such transformation is from UniLaSalle, a polytechnic university in France, where a ‘Living Lab’ was developed to provide students with sustainable development activities that are situated in real-world contexts (Fourati-Jamoussi et al., 2019).

Reviewer 3 Report

The paper is a finished piece and despite the presence of some disputable issues it is ready to be published. It is well designed and written. It has well prepared and presented methodology. It is well positioned versus the literature and up-to-date resources. the conclusion are based on the research results and it is confronted with existing concepts and literature. I propose to publish the paper.

Author Response

We (the authors) appreciate the time and effort you have made to review our manuscript. We have included your comments (bolded) and our response below.

The paper is a finished piece and despite the presence of some disputable issues it is ready to be published. It is well designed and written. It has well prepared and presented methodology. It is well positioned versus the literature and up-to-date resources. the conclusion are based on the research results and it is confronted with existing concepts and literature. I propose to publish the paper.

We appreciate your review and comments.

Reviewer 4 Report

The article is devoted to studying opinions of the engineering industry specialists and academia staff towards Engineering for Sustainable Development. This is a highly relevant and important field in modern science. The authors use qualitative methods to gather opinions of participants of the Engineering Global Development summit hosted by American Society of Mechanical Engineers.

The article is good and the conclusions are sound, but it can be enhanced by performing the following revisions:

1. The most important thing this article lacks is a definition of "engineer" and/or "engineering". Without it, how can we draw a border between engineering for sustainable society and professional engineers switching to other jobs or doing things in their spare time? This is important to draw the line between engineering for sustainable society and former engineers changing their profession to support sustainable society. A few questions that are worth answering:

1.1. Who is an engineer? A person with a relevant degree (what are the requirements for degree) whatever they do at their job? Or a person performing certain kind of jobs?

1.2. Can we consider people who perform "funding decision-making" or "policy development" engineers?

1.3. "political participation of engineers associated with political advocacy groups" - do they politically participate as engineers or as individuals (citizens)? Where is the border between engineering and personal life of an engineer? What about engineers who hold opposite political beliefs?

2. It will be good to add more demographics information (including the numbers) about the participants of the study besides simply stating that "Participants joined from all over the world, including Canada, India, Italy, Kenya, and the United Kingdom, with the majority participating from the United States. " Stats on participants countries, genders, ages and so on will allow the readers to put the study results in the proper context and increase the validity of the study which is based on the selected group of participants. It will also show how much the authors followed their own recommendations on diversity and inclusion.

3. The method of analysis of the obtained data should be explained in detail. For now, the authors just state that "Next, the two lead authors evaluated and synthesized the complete set of clustered data (i.e., all seven documents with consolidated coding) to identify emergent findings, including current trends, potential pathways forward, and recommendations for the engineering field" but there is no description of the procedure how this was done. This leaves open the questions if there were any selection bias when the findings were identified. We don't know the procedure by which the excerpts that are cited in the article were chosen and how typical they were among other answers. Providing more information on selection and the rate of agreement on these statements among the focus-group participants will allow to better evaluate the study validity.

English in the article is mostly good, but sometimes it is not very clear. E.g.,  in line 345"You need new ways to ensure you are incorporating the user's experience, needs, preferences, capacities..." did you really mean a single user or all users?

Author Response

We (the authors) appreciate the time and effort you have made to review our manuscript. We have included your comments (bolded) and our response below. We believe that your suggestions have strengthened our manuscript.

The article is devoted to studying opinions of the engineering industry specialists and academia staff towards Engineering for Sustainable Development. This is a highly relevant and important field in modern science. The authors use qualitative methods to gather opinions of participants of the Engineering Global Development summit hosted by American Society of Mechanical Engineers. The article is good and the conclusions are sound, but it can be enhanced by performing the following revisions:

  1. The most important thing this article lacks is a definition of "engineer" and/or "engineering". Without it, how can we draw a border between engineering for sustainable society and professional engineers switching to other jobs or doing things in their spare time? This is important to draw the line between engineering for sustainable society and former engineers changing their profession to support sustainable society. A few questions that are worth answering:

1.1. Who is an engineer? A person with a relevant degree (what are the requirements for degree) whatever they do at their job? Or a person performing certain kind of jobs? 1.2. Can we consider people who perform "funding decision-making" or "policy development" engineers?

We agree that our previous submission lacked a clear definition for “engineering” and the “engineer.” To address this, we have included additional definitions and background for our definition of engineering in our paper in our introduction:

“In recognition of these global challenges, efforts across various sectors and industries in the engineering field have sought to advance sustainability. The International Engineering Alliance defines engineering as a practice, body of knowledge, and set of techniques that can be purposefully applied to produce solutions of which the performance and impacts are predicted to the greatest extent possible (IEA, 2021). Many engineering organizations are moving away from disciplinary and siloed definitions, for example, the American Society of Mechanical Engineers (ASME) defines their mission as one that “promotes the art, science & practice of multidisciplinary engineering and allied sciences around the globe” (ASME, 2023). Engineering practice and knowledge can be applied to many different fields, including design, policy, business, and economics (MacDonald et al., 2022). In this manuscript, we employ these broad definitions of engineering—focusing on the large global field of technical applied sciences (ASME & Autodesk, 2022).”

1.3. "political participation of engineers associated with political advocacy groups" - do they politically participate as engineers or as individuals (citizens)? Where is the border between engineering and personal life of an engineer? What about engineers who hold opposite political beliefs?

Thank you for bringing these questions to our attention. In this section, we are referring to engineers participating in political conversations (e.g., regulations, standards, and policy) from their professional perspective (i.e., their area of technical expertise) not only as civilians. To clarify this point, we have included additional sentences in the section you have quoted (See Section 4.2.3: Recommendation 7):

“Most engineering associations and companies engage closely with government stakeholders, providing technical perspectives to conversations about regulations and policy changes. In this way, engineers have a great opportunity to participate in the political arena from not just as individual civilians, but as professionals, offering a much needed technical perspective.”

  1. It will be good to add more demographics information (including the numbers) about the participants of the study besides simply stating that "Participants joined from all over the world, including Canada, India, Italy, Kenya, and the United Kingdom, with the majority participating from the United States. " Stats on participants countries, genders, ages and so on will allow the readers to put the study results in the proper context and increase the validity of the study which is based on the selected group of participants. It will also show how much the authors followed their own recommendations on diversity and inclusion.

We agree that specific demographic information, such as age, gender, and nationality would strengthen the methods of this paper. Unfortunately, we did not collect this specific information in our participant surveys, and therefore did not collect participant consent to report this information.

  1. The method of analysis of the obtained data should be explained in detail. For now, the authors just state that "Next, the two lead authors evaluated and synthesized the complete set of clustered data (i.e., all seven documents with consolidated coding) to identify emergent findings, including current trends, potential pathways forward, and recommendations for the engineering field" but there is no description of the procedure how this was done. This leaves open the questions if there were any selection bias when the findings were identified. We don't know the procedure by which the excerpts that are cited in the article were chosen and how typical they were among other answers. Providing more information on selection and the rate of agreement on these statements among the focus-group participants will allow to better evaluate the study validity.

We agree that including more details in the data analysis methods will strengthen our paper. As such, we have included more detail on the thematic analysis process, which centered in team-based discussions as well as the process for reporting results. The additional details are presented in the Methods Section (See Section 2.4: Data Analysis and Member Checking).

“Our data analysis was grounded in the five objectives presented above. First, all notes from the summit and virtual whiteboards were examined independently by two individual evaluators, and participants' responses were grouped (i.e., deductively coded) into seven categories that mapped themes driven by our objectives: (1) Barriers for ESD careers, (2) Enablers for ESD careers, (3) Career pathway opportunities, (4) Skill development in ESD-trained engineers, (5) Shifting career pathways, (6) Changing mindsets, and (7) Other. For each of the seven focus groups, excerpts within each theme from the two evaluators were consolidated into a single document and organized by emerging themes (e.g., changing mindset: humility, shifting career pathways: policy careers). Next, three authors conducted semi-structured interviews with five other stakeholders who could not attend the summit for initial member checking and to assess the resonance and transferability of the preliminary results—between two to three authors conducted each of the interviews. Interview protocols followed the structure presented in Table 4.

Table 4. Interview protocol structure with key questions for member checking with additional stakeholders.

Introduction

● Interviewers present a summary of the ASME EGD Stakeholder Summit, including the overall goals and main session topics

Career pathways

● What skills and experiences are you needing in your organization? Any new positions developing?

● What barriers or enablers to sustainability careers do you see?

Trends at large

● What is needed for the sector to progress and achieve sustainability goals?

Next, we evaluated and synthesized the complete set of clustered data (i.e., all seven documents with consolidated coding categorized into emergent) to identify claims and key findings, including current trends, potential pathways forward, and recommendations for the engineering field. Using a shared word processing tool, we incorporated data from the member checking interviews with stakeholders to further fine-tune the claims being developed. Following recommendations by qualitative research scholars (Guest & McLellan, 2003), most of the authors participated in group discussions to negotiate the interactions between the raw data, emerging themes, and the overarching objectives. Findings (e.g., claims and themes) within each objective were reviewed, discussed, and iterated as part of a reflexive thematic analysis performed by the authors. 

The themes and claims within objectives 1-3 were then organized into three high-level categories to serve as the organization for reporting results in Section 3: 'Global drivers leading to mindset shifts,' 'Increased availability of professional pathways for ESD engineers,' and 'Broader skill sets for ESD engineers are valued.' When reporting the results, we carefully chose key findings from survey data or exemplary excerpts from our qualitative data to illustrate our analysis process and provide nuance to findings (Eldh et al., 2020). All presented excerpts are taken from the notetakers' detailed participation notes. To develop the discussion section (Section 4), we organized the key themes and claims identified for objective 4 to suggest opportunities for the field at large, which we organized into the following subsections: ‘Engineering for Sustainable Development: Terminology Recommendations,’ ‘ Emerging and recommended cultural shifts for field at large,’ ‘Incentives, metrics, and decision-making frameworks,’ and ‘Engineering education.’ Findings for objective 5 are underemphasized in this report and are being used to inform future ASME plans and programs.”

English in the article is mostly good, but sometimes it is not very clear. E.g.,  in line 345"You need new ways to ensure you are incorporating the user's experience, needs, preferences, capacities..." did you really mean a single user or all users?

Thank you for bringing this to our attention. We have revised the manuscript to fix this specific point, and have reviewed the entire manuscript to make improvements in the grammar.